# Investigating a Model-Agnostic and Imputation-Free Approach for Irregularly-Sampled Multivariate Time-Series Modeling

**Abhilash Neog[1]\***, **Arka Daw[2]**, **Sepideh Fatemi[1]**, **Medha Sawhney[1]**, **Aanish Pradhan[1]**, **Mary E. Lofton[1]**, **Bennett J. McAfee[3]**, **Adrienne Breef-Pilz[1]**, **Heather L. Wander[1]**, **Dexter W Howard[1]**, **Cayelan C. Carey[1]**, **Paul Hanson[3]**, **Anuj Karpatne[1]\***
*[1] Virginia Tech, [2] Oak Ridge National Lab, [3] University of Wisconsin-Madison*

**Reviewed on OpenReview:** *https://openreview.net/forum?id=HgJODMVAA3*

## Abstract

Modeling Irregularly-sampled and Multivariate Time Series (IMTS) is crucial across a variety of applications where different sets of variates may be missing at different time-steps due to sensor malfunctions or high data acquisition costs. Existing approaches for IMTS either consider a two-stage impute-then-model framework or involve specialized architectures specific to a particular model and task. We perform a series of experiments to derive insights about the performance of IMTS methods on a variety of semi-synthetic and real-world datasets for both classification and forecasting. We also introduce **Miss**ing Feature-aware **T**ime **S**eries **M**odeling (MissTSM) or MissTSM, a simple model-agnostic and imputation-free approach for IMTS modeling. We show that MissTSM shows competitive performance compared to other IMTS approaches, especially when the amount of missing values is large and the data lacks simplistic periodic structures–conditions common to real-world IMTS applications.

## 1 Introduction

Deep Learning for modeling multivariate Time-Series (MTS) is a rapidly growing field, with two major downstream tasks: forecasting and classification. Research (Dong et al., 2024; Nie et al., 2022a; Liu et al., 2023) in this field has been fueled by the availability of benchmark MTS datasets spanning diverse applications such as electric load forecasting and health monitoring containing fixed sets of variates regularly sampled over time. However, real-world MTS applications are plagued by missing values occurring over arbitrary sets of variates at every time-step (e.g. due to sensor malfunctions), resulting in Irregularly-sampled MTS (IMTS) datasets. IMTS modeling is particularly challenging because the misalignment of variates across time impairs transformer models that assume a fixed set of variates to be observed at every time-step

A common approach for IMTS modeling is to use a two-step framework where we first use imputation methods (Ahn et al., 2022; Batista et al., 2002) to fill in missing values based on observed data, followed by feeding the imputed time-series to an MTS model (see Figure 1). Note that the choice of imputation method is agnostic to the MTS model, making it "model-agnostic." However, the effectiveness of this framework relies on the quality of performed imputation, which can degrade if the time-series lacks periodic structure or if the imputation method is overly simplistic. Imputation can also introduce artificial patterns or artifacts into the data, which MTS models may interpret as genuine trends or observations. Moreover, deep learning-based imputation methods require training, which adds to the overall computational cost of IMTS modeling.

Imputation-free approaches for IMTS have also been developed in recent literature (Che et al., 2018; Rubanova et al., 2019), that involve specialized architectures to handle missing values in time-series for specific

---

\*Corresponding author: abhilash22@vt.edu, karpatne@vt.edu
Code available at: **https://github.com/abhilash-neog/SparseTimeSeriesModeling**

downstream tasks such as classification (see Figure 1) However, these approaches have been empirically shown to struggle with capturing long-term temporal dependencies that are central to the problem of forecasting, are difficult to parallelize, and incur high computational costs. Furthermore, existing imputation-free IMTS approaches are not model-agnostic, i.e., they have been developed as specialized model architectures that cannot be used as a generic wrapper with latest advances in MTS models, restricting their adaptability and performance.

Given the complementary strengths and weaknesses of existing approaches in IMTS modeling, we ask the following question - *can we develop a model-agnostic and imputation-free approach for IMTS modeling that can be used in a variety of downstream tasks (e.g., forecasting and classification)*? To address this question, we analyze the prevailing strategies for converting time-series into tokens in existing transformer-based MTS models. There are two primary ways for converting time series into tokens: (a) treating each time step (all variates) as a token, or (b) considering each variate (all time steps) as a token (Liu et al., 2023). While these strategies work well for regular MTS data, they are not suited to handle IMTS data because we may be missing some variates at a time step or some time-steps for a variate, making the computation of tokens infeasible. In contrast, we explore a different perspective for creating time-series tokens: independently embedding each combination of time-step and variate as a token. Time-variate combinations with missing values

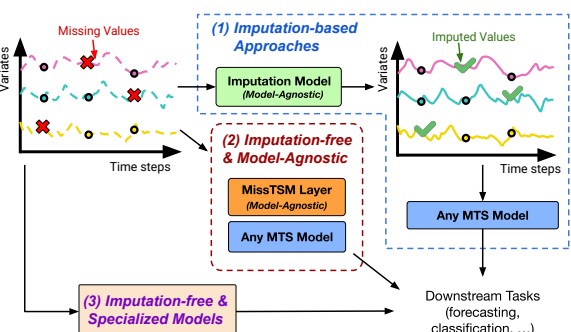

Figure 1: We investigate the relative importance of three categories of approaches for modeling irregular and multivariate time-series: (1) imputation-based approaches, (2) model-agnostic and imputation-free approaches (proposed MissTSM layer), and (3) imputation-free approaches involving specialized architectures.

can then be handled using masked cross-attention without performing any explicit imputation. Building on this intuition, we introduce MissTSM, a simple model-agnostic and imputation-free approach for IMTS modeling, designed as a "plug-and-play" layer that can be integrated into any backbone MTS model to handle IMTS data. The advantage of such an approach is that it *(a) does not introduce any imputation artifacts*, and *(b) can act as a wrapper around any MTS model.*

In this work, we make the following contributions: **(1)** We introduce MissTSM, a model-agnostic and imputation-free approach; **(2)** We conduct a comprehensive experimental study on a variety of datasets for both classification and forecasting tasks, using synthetic masking techniques as well as real-world occurrence of missing values. This study investigates *(a) sensitivity of imputation-based frameworks on the choice of imputation technique and the nature of missing values*, and *(b) the performance of IMTS approaches as the fraction of missing values varies*; **(3)** We demonstrate that MissTSM achieves competitive performance compared to other IMTS approaches, especially when the amount of missing values is large and the data lacks simplistic periodic structures - conditions common to real-world IMTS applications.

## 2 Related Works

**Time-series Forecasting.** With the introduction of attention mechanisms via transformer models (Vaswani et al., 2017), a number of transformer-based time-series models have been developed in the last few years (Wu et al., 2021; Nie et al., 2022b; Dong et al., 2024; Liu et al., 2023). While Transformer-based models have shown great promise, recently there has been a strong interest in exploring the use of simple linear models for time-series forecasting as well (Zeng et al., 2023; Ekambaram et al., 2023). In addition, with the rise of self-supervised learning-based models such as masked auto-encoders (MAEs) (He et al., 2022), a new category of MAE-style time-series models have emerged (Dong et al., 2024) that have received a lot of recent interest owing to their ability of learning both low-level and high-level representations for varied downstream tasks such as forecasting and classification. However, while these methods can deal with missing values in the temporal domain, they are unable to handle missing values across both variates and time.

**Imputation Methods.** Traditionally, most imputation techniques for handling missing values in time-series have been based on statistical approaches (Fung, 2006; Batista et al., 2002; Dempster et al., 1977; Mnih & Salakhutdinov, 2007). In recent years, there is a growing trend to use deep learning methods for time-series imputation, such as SAITS (Du et al., 2023), CSDI (Tashiro et al., 2021a), GAIN (Yoon et al., 2018a), and BRITS (Cao et al., 2018). Imputation techniques can be broadly classified into two classes: those that leverage cross-channel correlations (Batista et al., 2002; Acuna & Rodriguez, 2004) and those that exploit temporal dynamics (Box et al., 2015). Recently, deep learning-based approaches for imputation have been developed (Tashiro et al., 2021b; Cini et al., 2021; Liu et al., 2019; Cao et al., 2018; Du et al., 2023), which can jointly learn the temporal dynamics with cross-channel correlations. These methods, however, rely on a single entangled representation (or hidden state) to model nonlinear dynamics (Woo et al., 2022) which can be a limitation in capturing the multifaceted nature of time-series. Matrix factorization based techniques (Liu et al., 2022) have also been proposed that offer disentangled temporal representations, enhancing the ability to differentiate and model distinct temporal features. While these deep learning-based models are highly efficient during inference, they require additional training time, which add to the already large time complexity of MTS models.

**Imputation-free IMTS Models.** In the last decade, there has been a significant growth of models and architectures for learning from IMTS data. Some of the simpler approaches to deal with IMTS data involve working with fixed temporal discretization (Marlin et al., 2012; Lipton et al., 2016). The primary drawback with these approaches is that they make ad-hoc choices in terms of discretization window width and aggregation functions within the windows (Shukla & Marlin, 2020). A popular set of approaches for handling IMTS data are recurrence-based approaches, which includes RNN-based methods such as GRU-D (Che et al., 2018). However, GRU-D has limited scalability to long sequences. Other recurrence-based approaches based on Ordinary Differential Equations (ODE) (Chen et al., 2018; Rubanova et al., 2019) provide an effective solution in modeling the continuous time semantics. These methods are however significantly slow and memory-intensive, as they constantly need to apply the ODE solver and solving ODEs require numerical integration, thus making it impractical for long-term forecasting and large datasets.

Transformer-based methods such as ContiFormer (Chen et al., 2023) and mTAN (Shukla & Marlin, 2021) addresses these limitations by explicitly integrating the modeling abilities of Neural ODEs into the attention mechanism and introducing continuous time attention mechanism that learns time embeddings dynamically, respectively. However, despite ContiFormer being a principled and effective approach, solving an ODE for each key and value incurs a high computational cost. Also, while the runtime speed of mTAN is relatively faster, it is however, inherently optimized toward interpolating missing values by learning representations at fixed set of reference points, thus limiting it's extrapolation or forecasting ability.

This is another limitation of IMTS approaches—their evaluation is mostly limited to a single task, most often to time series classification, thus limiting their applicability. ContiFormer performs evaluation on forecasting tasks, however, they consider regular and clean benchmark time-series datasets in their evaluation. Another limitation in terms of evaluation is that the prior works primarily focus on other IMTS models for comparison, completely ignoring the two-stage imputation approach, which is a more common and practical way of dealing with missing-value data. Our work aims to solve these issues by providing a comprehensive comparison against both existing imputation-free and two-stage imputation-based approaches, and proposing a model-agnostic transformation-allowing any task-specific SOTA model to be applied on any irregularly sampled time-series data with minimal data transformations.

## 3 Proposed Missing Feature Time-Series Modeling (MissTSM) Framework

### 3.1 Notations and Problem Formulations

Let us represent a multivariate time-series as $\mathbf{X} \in \mathbb{R}^{T \times N}$, where $T$ is the number of time-steps, and $N$ is the dimensionality (number of variates) of the time-series. We assume a subset of variates (or features) to be missing at some time-steps of $\mathbf{X}$, represented in the form of a missing-value mask $\mathcal{M} \in [0, 1]^{T \times N}$, where $\mathcal{M}_{(t,d)}$ represents the value of the mask at $t$-th time-step and $d$-th dimension. $\mathcal{M}_{(t,d)} = 1$ denotes that the corresponding value in $\mathbf{X}_{(t,d)}$ is missing, while $\mathcal{M}_{(t,d)} = 0$ denotes that $\mathbf{X}_{(t,d)}$ is observed. Furthermore, let us

denote $\mathbf{X}_{(t,:)} \in \mathbb{R}^N$ as the multiple variates of the time-series at a particular time-step $t$, and $\mathbf{X}_{(:,d)} \in \mathbb{R}^T$ as the uni-variate time-series for the variate $d$. In this paper, we consider two downstream tasks for time-series modeling: forecasting and classification. For forecasting, the goal is to predict the future $S$ time-steps of $\mathbf{X}$ represented as $\mathbf{Y} \in \mathbb{R}^{S \times N}$. Alternatively, for time-series classification, the goal is to predict output labels $\mathbf{Y} \in \{1, 2, ..., C\}$ given $\mathbf{X}$, where $C$ is the number of classes.

### 3.2 Learning Embeddings for Time-Series with Missing Features

**Limitations of Existing Transformer Methods:** The first step in time-series modeling using transformer-based architectures is to learn an embedding of the time-series $\mathbf{X}$ that can be sent to the transformer encoder. Traditionally, this is done using an Embedding layer (typically implemented using a multi-layered perceptron) as $\texttt{Embedding} : \mathbb{R}^N \mapsto \mathbb{R}^D$ that maps $\mathbf{X} \in \mathbb{R}^{T \times N}$ to the embedding $\mathbf{H} \in \mathbb{R}^{T \times D}$, where $D$ is the embedding dimension. The Embedding layer operates on every time-step independently such that the set of variates observed at time-step $t$, $\mathbf{X}_{(t,:)}$, is considered as a single token and mapped to the embedding vector $\mathbf{h}_t \in \mathbb{R}^D$ as $\mathbf{h}_t = \texttt{Embedding}(\mathbf{X}_{(t,:)})$ (see Figure 2(a)).

An alternate embedding scheme was recently introduced in the framework of inverted Transformer (iTransformer) (Liu et al., 2023), where the uni-variate time-series for the $d$-th variate, $\mathbf{X}_{(:,d)}$, is considered as a single token and mapped to the embedding vector: $\mathbf{h}_d = \texttt{Embedding}(\mathbf{X}_{(:,d)})$ (see Figure 2(b)).

While both these embedding schemes have their unique advantages, they are unfit to handle time-series with arbitrary sets of missing values at every time-step. In particular, the input tokens to the Embedding layer of Transformer or iTransformer requires all components of $\mathbf{X}_{(t,:)}$ or $\mathbf{X}_{(:,d)}$ to be observed, respectively. If any of the components in these tokens are missing, we will not be able to compute their embeddings and thus will have to discard either the time-step or the variate, leading to loss of information.

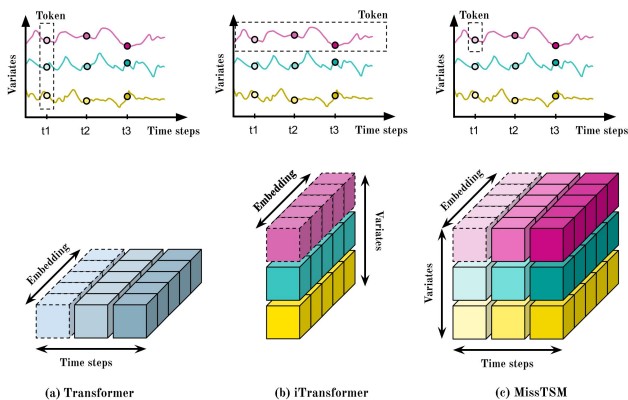

Figure 2: Schematic of the Time-Feature Independent (TFI) Embedding of MissTSM that learns a different embedding for every combination of time-step and variate, in contrast to the time-only embeddings of Transformer (Vaswani et al., 2017) and the variate-only embeddings of iTransformers (Liu et al., 2023).

**Time-Feature Independent (TFI) Embedding:** To address this challenge as well as to utilize inter-variate interactions similar to Wei et al. (2023), we consider a *Time-Feature Independent (TFI) Embedding* scheme for time-series with missing features, where the value at each combination of time-step $t$ and variate $d$ is considered as a single token $\mathbf{X}_{(t,d)}$, and is independently mapped to an embedding using $\texttt{TFIEmbedding} : \mathbb{R} \mapsto \mathbb{R}^D$ as follows:

$$\mathbf{h}_{(t,d)} = \texttt{TFIEmbedding}(\mathbf{X}_{(t,d)}) \tag{1}$$

In other words, the $\texttt{TFIEmbedding}$ Layer (which is a simple MLP layer) maps $\mathbf{X} \in \mathbb{R}^{T \times N}$ into the TFI embedding $\mathbf{H}^{\text{TFI}} \in \mathbb{R}^{T \times N \times D}$ (see Figure 2(c)). The $\texttt{TFIEmbedding}$ is applied only on tokens $\mathbf{X}_{(t,d)}$ that are observed (for missing tokens, i.e., $\mathcal{M}_{(t,d)} = 1$, we generate a dummy embedding that gets masked out in the MFAA layer). The advantage of such an approach is that even if a particular value in the time-series is missing, other observed values in the time-series can be embedded *"independently"* without being affected by the missing values. Moreover, it allows the MFAA layer to leverage the high-dimensional embeddings to store richer representations bringing in the context of time and variate by computing masked cross-attention among the observed features at a time-step to account for the missing features.

**2D Positional Encodings:** We add Positional Encoding vectors $\mathbf{PE}$ to the TFI embedding $\mathbf{H}^{\text{TFI}}$ to obtain positionally-encoded embeddings, $\mathbf{Z} = \mathbf{PE} + \mathbf{H}^{\text{TFI}}$. Since TFI embeddings treat every time-feature

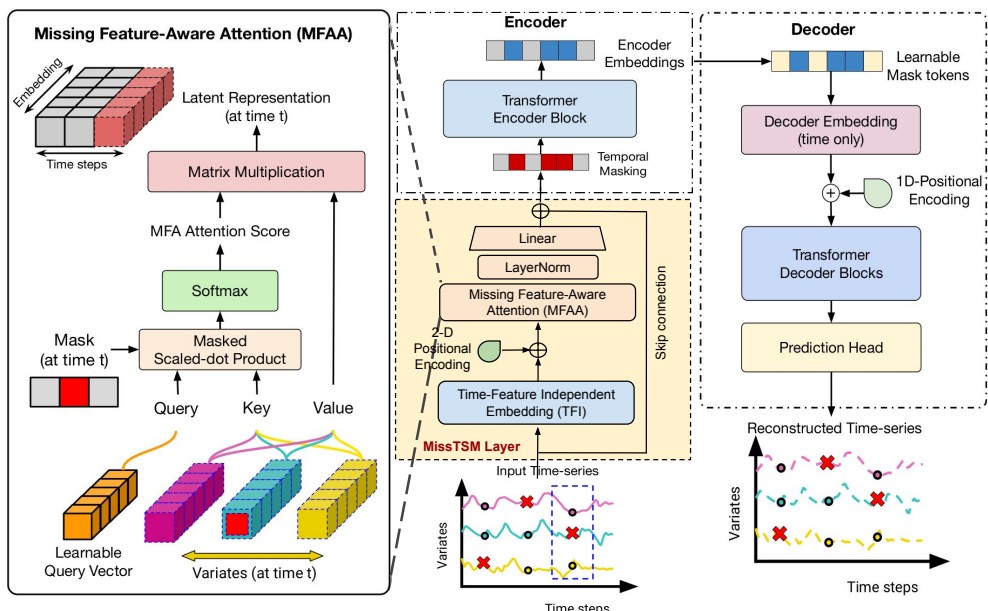

Figure 3: Overview of the MissTSM layer integrated within the Masked Auto-Encoder framework (Li et al., 2023). A zoomed-in view of the MFAA is shown on the left.

combination as a token, we use a 2D-positional encoding scheme defined as follows:

$$\text{PE}(t, d, 2i) = \sin\left(\frac{t}{10000^{(4i/D)}}\right),$$
$$\text{PE}(t, d, 2i+1) = \cos\left(\frac{t}{10000^{(4i/D)}}\right) \tag{2}$$

$$\text{PE}(t, d, 2j + D/2) = \sin\left(\frac{d}{10000^{(4j/D)}}\right),$$
$$\text{PE}(t, d, 2j+1 + D/2) = \cos\left(\frac{d}{10000^{(4j/D)}}\right) \tag{3}$$

where $t$ is the time-step, $d$ is the feature, and $i, j \in [0, D/4)$ are integers.

### 3.3 Missing Feature-Aware Attention (MFAA)

The MFAA Layer illustrated in Figure 3 leverage the power of *"masked-attention"* for learning latent representations at every time-step using partially observed features. MFAA works by computing *attention scores* based on the partially observed features at a time-step $t$, which are then used to perform a weighted sum of observed features to obtain the latent representation $\mathbf{L}_t$. As shown in Figure 3, these latent representations are projected back using a linear layer, to the original input shape before being fed into the downstream model (here, the encoder-decoder based self-supervised learning framework). Note that MFAA is not designed for long-term temporal modeling. Its primary role is to handle missing values at a time-step by using the observed variates at the same time-step as context. It relies on the subsequent backbone to model long-range temporal dynamics MFAA performs a masked cross-attention using a learnable query vector and observed data as keys and values. This separation of roles is inspired by similar architectures in multi-modal grounding, for example, in Carion et al. (2020), where learnable object queries serve as abstract object representations to focus on distinct objects in an image without requiring predefined region proposals, enabling set-based prediction. Similarly, in our setting, the learnable queries capture the interactions among variates independent of time, enabling the model to attend to the most informative aspects of observed variates at any time-step fed through keys and values. This intuition aligns with the query-based mechanism in mTAN (Shukla & Marlin, 2021), which introduces a structured way to aggregate information over observed time-series data.

However, there's is a key difference in the nature of the query - while mTAN uses discrete reference points on a fixed temporal grid to achieve this, our single learnable query generalizes across variates at every time step, allowing for a more flexible representation of feature interactions. Another key difference is in the architecture flow - mTAN first performs a "temporal-first, variate-independent interpolation" to perform temporal attention for each feature to create a regular sequence, then performs a linear combination to generate a single per timestep representation. In contrast, MissTSM is "variate-first", with the MFAA layer performing cross-variate attention to create a representation $L_t$ at each timestep. Then, the backbone model learns the temporal relationships from this sequence of summaries. Also note that our goal is not to propose a new attention mechanism, rather, our novelty lies in the novel adaptation of masked cross-attention for effectively modeling IMTS data. To the best of our knowledge, this novel adaptation of masked cross-attention for IMTS modeling has not been tried before in any previous works.

**Mathematical Formulation:** To obtain attention scores from partially observed features at a time-step, we apply a masked scaled-dot product operation followed by a softmax operation described as follows. We first define a learnable query vector $\mathbf{Q} \in \mathbb{R}^{1 \times D}$ which is independent of the variates and time-steps. The positionally-encoded embeddings at time-step $t$, $\mathbf{Z}_{(t,:)}$, are used as key and value inputs in the MFAA Layer. Specifically, The query, key, and value vectors are defined using linear projections as follows: $\hat{\mathbf{Q}} = \mathbf{Q}\mathbf{W}^{\mathbf{Q}}$, $\hat{\mathbf{K}}_t = \mathbf{Z}_{(t,:)}\mathbf{W}^{\mathbf{K}}$, $\hat{\mathbf{V}}_t = \mathbf{Z}_{(t,:)}\mathbf{W}^{\mathbf{V}}$. Here, $\hat{\mathbf{Q}} \in \mathbb{R}^{1 \times d_k}$ and $\hat{\mathbf{K}}_t, \hat{\mathbf{V}}_t \in \mathbb{R}^{N \times d_k}$, where $d_k$ is the dimension of the vectors after linear projection. The linear projection matrices for the query, key, and values are defined as: $\mathbf{W}^{\mathbf{Q}}, \mathbf{W}^{\mathbf{K}}, \mathbf{W}^{\mathbf{V}} \in \mathbb{R}^{D \times d_k}$ respectively. Note that the key $\hat{\mathbf{K}}_t$ and value $\hat{\mathbf{V}}_t$ vectors depend on the time-step $t$, while the query vector doesn't change with time. We then define the Missing Feature-Aware Attention Score at a given time-step $t$ as a masked scalar dot-product of the query and key vector followed by normalization of the scores using a Softmax operation, formally defined as follows:

$$\mathbf{A}_t = \texttt{MFAAScore}(\hat{\mathbf{Q}}, \hat{\mathbf{K}}_t, \mathcal{M}_{(t,:)})$$
$$= \texttt{Softmax}\left(\frac{\hat{\mathbf{Q}}\hat{\mathbf{K}}_t^\top}{\sqrt{d_k}} + \eta\,\mathcal{M}_{(t,:)}\right) \tag{4}$$

where $\mathbf{A}_t \in \mathbb{R}^N$ is the MFAA Score vector of size $N$ corresponding to the $N$ variates, and $\eta \to -\infty$ is a large negative bias. The negative bias term $\eta$ forces the masked-elements that correspond to the missing variates in the time-series to have an attention score of zero. Thus, by definition, the $i$-th element of the MFAA Score $\mathbf{A}_{(t,i)} \neq 0 \implies \mathcal{M}_{(t,:)} = 0$. We compute the latent representation $\mathbf{L}_t$ as a weighted sum of the MFAA score $\mathbf{A}_t$ and the Value vector $\hat{\mathbf{V}}_t$ as follows:

$$\mathbf{L}_t = \texttt{MFAA}(\mathbf{A}_t, \hat{\mathbf{V}}_t) = \mathbf{A}_t\hat{\mathbf{V}}_t \in \mathbb{R}^{d_k} \tag{5}$$

Similar to multi-head attention used in traditional transformers, we extend MFAA to multiple heads as follows:

$$\texttt{MultiHeadMFAA}(\mathbf{Q}, \mathbf{Z}_{(t,:)}, \mathcal{M}_{(t,:)})$$
$$= \text{Concat}(\mathbf{L}_t^0, \mathbf{L}_t^1, \ldots, \mathbf{L}_t^{h-1}) \cdot \mathbf{W}^O \tag{6}$$

where $h$ is the number of heads, $\mathbf{W}^{\mathbf{0}} \in \mathbb{R}^{hd_k \times D_o}$, $\mathbf{L}_t^i$ is the latent representation obtained from the $i$-th `MHAA` Layer, and $D_o$ is the output-dimension of the `MultiHeadMFAA` Layer.

### 3.4 Putting Everything Together: Plugging MissTSM with any MTS Model

Figure 3 shows the overall framework of a Masked Auto-Encoder (MAE) (He et al., 2022) based time-series model integrated with MissTSM. For an input time-series $\mathbf{X}$, we apply the TFI embedding layer followed by the MFAA layer to learn a latent representation for every time-step. The latent representations are then projected back to the original input shape to be fed into the downstream model. In this work, we opted for a MAE-based time-series model as the default downstream or base model, primarily due to its recent success in time-series modeling and its ability to perform both time-series forecasting and classification tasks. Furthermore, out of the several state-of-the-art masked time-series modeling techniques, we intentionally chose the simplest variation of MAE, namely Ti-MAE (Li et al., 2023), to highlight the effectiveness of TFI and MFAA layers in handling missing values.

# 4 Experimental Setup

**Baselines:** We benchmark against two categories of models. For MTS, we consider SimMTM (Dong et al., 2024), PatchTST (Nie et al., 2022b), AutoFormer (Wu et al., 2021), DLinear (Zeng et al., 2023), and iTransformer (Liu et al., 2023). Imputation strategies used are, $2^{\text{nd}}$-order spline interpolation (McKinley & Levine, 1998), k-Nearest Neighbor (Tan et al., 2019), and SAITS (Du et al., 2023) and BRITS (Cao et al., 2018). For IMTS, we evaluate GRU-D (Che et al., 2018), Latent ODE (Rubanova et al., 2019), SeFT (Horn et al., 2020), mTAND (Shukla & Marlin, 2021), Raindrop (Zhang et al., 2021), and MTGNN (Wu et al., 2020). Baseline choice is aligned with the task each model was originally designed for.

**Datasets:** We considered three popular time-series forecasting datasets: ETTh2, ETTm2 (Zhou et al., 2021) and Weather (Weather, 2021). For classification, we considered three real-world datasets, namely, Epilepsy (Andrzejak et al., 2001), EMG (Goldberger et al., 2000a), and Gesture (Liu et al., 2009). We follow the same evaluation setups as proposed in TF-C (Zhang et al., 2022). To simulate varying scenarios of missing values appearing in real-world time-series datasets, we adopt two synthetic masking schemes that we apply on these benchmark datasets, namely missing completely at random (MCAR) masking and periodic masking. Furthermore, we compared our performance on five real-world datasets: PhysioNet-2012 (Silva et al., 2012), P12 (Goldberger et al., 2000b) and P19 (Reyna et al., 2020) for health monitoring; Falling Creek Reservoir (FCR) dataset for modeling lake water quality, and Lake Mendota from the North Temperate Lakes Long-Term Ecological Research program (NTL-LTER; Magnuson et al., 2024) also for modeling lakes. See Appendix for more details.

# 5 Results and Discussions

Here, we discuss our findings with respect to imputation-based vs. imputation free methods, and model-agnostic vs. specialized methods across a variety of datasets, tasks, and missing value settings.

## 5.1 Imputation-based vs. Imputation-free

### 5.1.1 Impact of Missing Data Fractions.

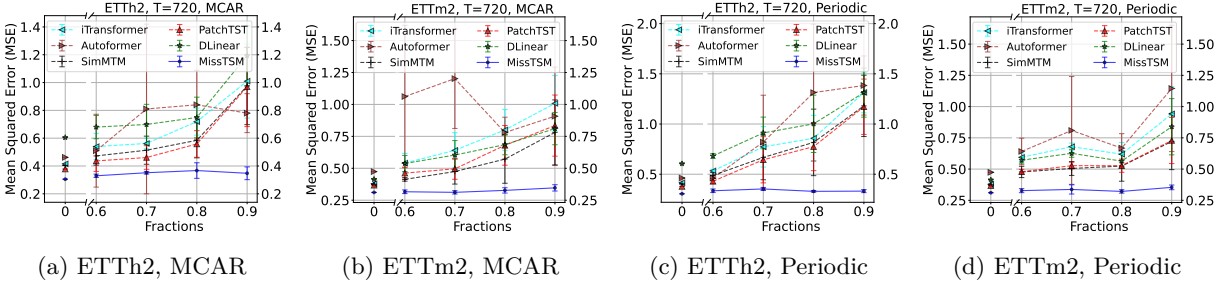

Figure 4: Performance comparison against different TS Baselines imputed with SAITS, across different missing data fractions.

To understand the effect of varying masking fractions on the forecasting performance, we consider five forecasting models trained on SAITS-imputed data as the set of imputation-based baselines. We compare their results with MissTSM integrated within the MAE framework as an imputation-free approach. Figure 4 shows variations in the Mean Squared Error (MSE) as we increase the missing value fraction in MCAR and periodic masking scheme from 0.6 to 0.9 for forecasting horizon $T = 720$ on two ETT datasets. We can see that, on average, as we increase the amount of missing values in the data, imputation-based baselines and MissTSM show an increasing trend in MSE. This is expected as larger missing value fractions starve IMTS models with greater amount of information degrading their performance. However, the rise in MSE of MissTSM with missing value fractions is much less pronounced than imputation-based baselines consistently across the two datasets and synthetic masking schemes (MCAR and Periodic Masking). Further, note that

MissTSM shows smaller standard deviations compared to the large and varying standard deviations of the imputation-based approaches (w.r.t the increasing missingness). These results suggest that imputation-based frameworks struggle when the amount of missing values is high, possibly due to the poor performance of imputation methods when the number of observations is small.

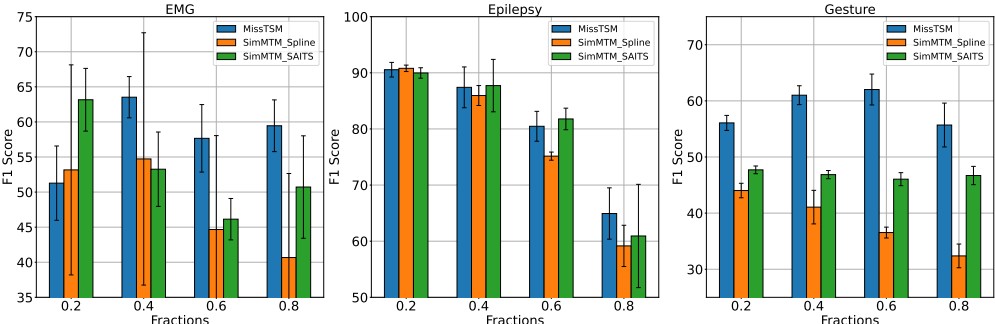

Figure 5: Classification F1 scores on three datasets, EMG, Epilepsy, and Gesture. Masking fractions considered: 0.2, 0.4, 0.6, 0.8.

We conduct a similar study to understand the impact of missing data fractions on classification tasks with MCAR masking scheme (see Figure 5). Similar to forecasting, we see, on average, a gradual decrease in the F1 scores with increasing missing fractions of the imputation-based approaches. We also observe a high range of variability in the Spline-imputed baselines, which suggests that the polynomial order of spline imputation can be further fine-tuned specific to the data. On the other hand, MissTSM shows consistently strong performance across all the three datasets.

### 5.1.2   Impact of the Nature of Missing Values.

To understand the performance of imputation-based and imputation-free approaches under varying conditions of missing data, we compared their results across the two synthetic masking patterns: MCAR and Periodic Masking. From Fig. 6, we can observe that for the ETTh2 dataset, models perform consistently better under random masking compared to periodic. We can also see that the performance difference between MCAR and Periodic masking is, on an average, higher for SAITS-imputed models compared to Spline. This suggests that the hyper-parameters of SAITS can be further fine-tuned on the Periodic dataset, which is relatively easier for Spline to model. Additionally, the performance under MCAR and Periodic missingness on Weather dataset is comparatively similar, which hint towards high seasonality within the weather dataset, thus helping the imputation-based baselines on this dataset.

### 5.1.3   Impact of Imputation Methods.

The choice of imputation method dictates the overall performance of imputation-based frameworks. In Figure 7a, we compare four imputation techniques: Spline, kNN, BRITS, and SAITS, paired with two MTS models (iTransformer and PatchTST) at a forecasting horizon of $T = 720$ and 70% missing data fraction. Model performance on BRITS-imputed data is relatively poor, whereas models trained on SAITS-imputed data performs relatively good. This difference in performance indicates the impact of imputation models on downstream tasks within imputation-based frameworks. Notably, MissTSM-based imputation-free model achieves relatively low MSE scores compared to most imputation-based frameworks.

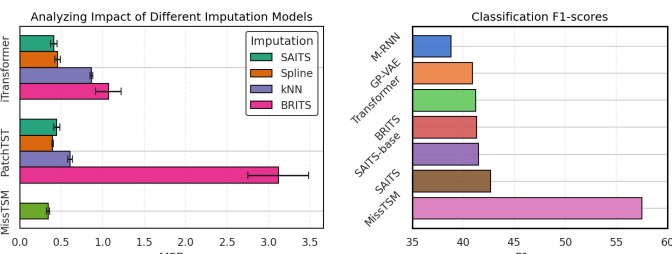

(a) Varying imputation models. Performance on ETTh2.

(b) Classification results of imputation models on PhysioNet.

Figure 7: Comparison of MSE (lower is better) and F1-score across imputation methods.

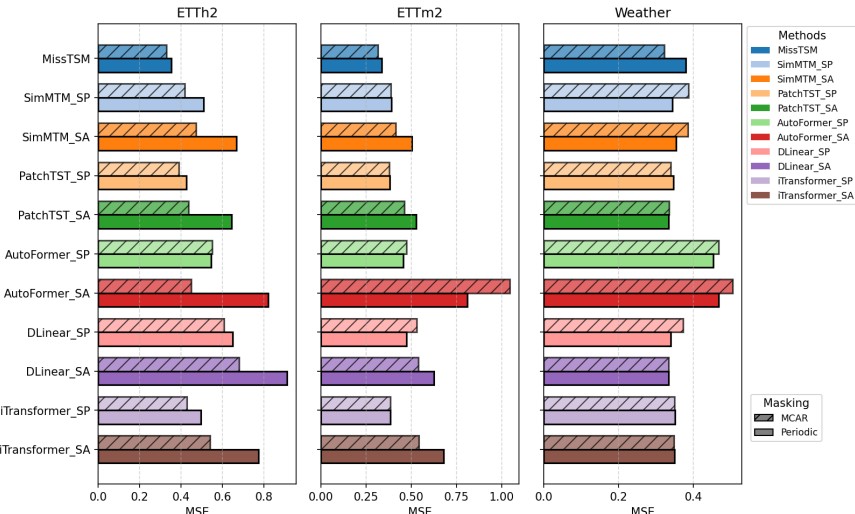

Figure 6: Comparison of different masking methods (70% missing fraction): MCAR vs. Periodic Masking for ETTh2, ETTm2, and Weather datasets. SA stands for SAITS and SP stands for Spline. Lower MSE is better

In Figure 7b, we compare MissTSM with six imputation baselines - M-RNN (Yoon et al., 2018b), GP-VAE (Fortuin et al., 2020), BRITS (Cao et al., 2018), Transformer (Vaswani et al., 2017), and SAITS (Du et al., 2023) - on a popular real-world classification dataset, PhysioNet (Silva et al., 2012) following the same evaluation setup as proposed in (Du et al., 2023). MissTSM achieves an impressive F1-score of 57.84%, representing an approximately 15% improvement over the best-performing model (trained on SAITS imputed data). This substantial performance gain on a real-world dataset with missing values highlights the potential of imputation-free or single-stage approaches compared to imputation-based approaches.

## 5.2 Comparing Model-Agnostic vs. Specialized Models

### 5.2.1 Analyzing MissTSM on IMTS Classification and Forecasting

We evaluate MissTSM on both classification and forecasting tasks for irregular multivariate time series. To illustrate the generality of our approach, we study two case models: (i) GRU-D, a specialized classifier for irregularly sampled data, and (ii) Latent ODE, a continuous-time generative model not originally designed for forecasting but adapted here to a long-term prediction setting. These case studies emphasize how specialized methods struggle when moved beyond their intended use, underscoring the value of model-agnostic approaches.

**IMTS Classification**. We conduct experiments on the IMTS classification task using the P12 (Goldberger et al., 2000b) and P19 (Reyna et al., 2020) datasets, following the same evaluation protocol as Luo et al. (2025). We report the baseline results for the considered models directly from Luo et al. (2025). Table 1 highlights the strong potential of model-agnostic approaches; integrating the MissTSM layer, can achieve performance on par with or exceeding that of several well-known IMTS models.

**Comparing MissTSM with GRU-D on Classification.** To analyze the potential of model-agnostic approaches we apply the same MissTSM-integrated MAE model on synthetically masked (80%) classification datasets and compare against GRU-D. From Table 2, we observe that while GRU-D is a specialized model for IMTS data, the proposed model-agnostic still outperforms it significantly. Please refer to the Appendix for more implementation details.

**Comparing MissTSM with Latent ODE on ETTh2.** As discussed above, specialized IMTS models cannot be easily adapted to a different task. To analyze this further, we adapt the Latent ODE model (with ODE-RNN encoder) for a long-term forecasting problem and compare it against our model-agnostic approach. We consider a simple setup with 336 context length and 96 prediction length under MCAR masking with varying fractions. From Table 3, we see that Latent ODE struggles to perform long-sequence modeling, with

Table 1: Performance comparison on P19 and P12. Best in bold, second-best underlined.

| Methods | P19 | | P12 | |
|---|---|---|---|---|
| | **AUROC** | **AUPRC** | **AUROC** | **AUPRC** |
| GRU-D | $\underline{88.7}_{1.2}$ | $\underline{56.2}_{2.3}$ | $79.6_{0.6}$ | $41.7_{1.8}$ |
| ODE-RNN | $87.1_{1.0}$ | $52.6_{3.2}$ | $78.8_{0.6}$ | $37.4_{2.6}$ |
| SeFT | $84.0_{0.3}$ | $49.3_{0.5}$ | $78.1_{0.5}$ | $35.9_{0.8}$ |
| mTAND | $82.9_{0.9}$ | $32.2_{1.5}$ | $\mathbf{85.3}_{0.3}$ | $\mathbf{49.3}_{1.0}$ |
| Raindrop | $87.6_{2.7}$ | $\mathbf{61.1}_{1.4}$ | $82.0_{0.6}$ | $42.7_{1.7}$ |
| MTGNN | $88.5_{1.0}$ | $55.8_{1.5}$ | $82.1_{1.5}$ | $41.8_{2.1}$ |
| MissTSM | $\mathbf{88.8}_{1.3}$ | $\underline{56.5}_{1.2}$ | $\underline{82.2}_{0.5}$ | $\underline{43.8}_{1.1}$ |

Table 2: Comparing (F1 scores) MissTSM approach against GRU-D for classification datasets.

| Dataset | GRU-D | MissTSM |
|---|---|---|
| Epilepsy | 6.52% | 64.9% |
| Gesture | 3.16% | 55.70% |
| EMG | 2.78% | 59.45% |

Table 3: Comparing (MSE values) MissTSM with Latent ODE adapted for forecasting

| Fraction | Latent ODE | MissTSM |
|---|---|---|
| 60% | 4.25 | 0.243 |
| 70% | 3.181 | 0.250 |
| 80% | 2.543 | 0.264 |
| 90% | 2.624 | 0.316 |

significantly high MSE values. Moreover, ODE-based methods incur considerable computational costs, which grow even more pronounced for long-term modeling.

### 5.2.2 Analyzing Model-Agnostic Nature of MissTSM.

To further analyze model-agnostic capability of the proposed approach we integrate MissTSM with other MTS models like PatchTST and iTransformer. Tables 4 and 5 show competitive performance of MissTSM integrated with PatchTST, revealing potential for plugging MissTSM with advanced MTS models for improved performance on downstream tasks even in the presence of missing values with minimal change to the MTS model architecture. Please refer to appendix for additional results.

Table 4: MSE (mean$_{std}$) for PatchTST with MissTSM under 60% masking.

| Dataset | Horizon Window | PatchTST + MissTSM | PatchTST + SAITS | PatchTST + Spline |
|---|---|---|---|---|
| ETTh2 | 96 | $\mathbf{0.317}_{0.004}$ | $0.503_{0.013}$ | $\underline{0.324}_{0.013}$ |
| | 192 | $\mathbf{0.377}_{0.009}$ | $0.512_{0.011}$ | $\underline{0.399}_{0.017}$ |
| | 336 | $\mathbf{0.380}_{0.011}$ | $\underline{0.410}_{0.012}$ | $0.431_{0.005}$ |
| | 720 | $0.514_{0.033}$ | $\mathbf{0.411}_{0.002}$ | $\underline{0.436}_{0.017}$ |
| ETTm2 | 96 | $\underline{0.202}_{0.005}$ | $0.322_{0.045}$ | $\mathbf{0.169}_{0.000}$ |
| | 192 | $\underline{0.261}_{0.002}$ | $0.359_{0.036}$ | $\mathbf{0.227}_{0.000}$ |
| | 336 | $\underline{0.313}_{0.001}$ | $0.408_{0.043}$ | $\mathbf{0.285}_{0.001}$ |
| | 720 | $\underline{0.420}_{0.027}$ | $0.459_{0.035}$ | $\mathbf{0.376}_{0.001}$ |
| Weather | 96 | $\underline{0.206}_{0.014}$ | $\mathbf{0.169}_{0.001}$ | $0.270_{0.110}$ |
| | 192 | $\underline{0.276}_{0.027}$ | $\mathbf{0.212}_{0.000}$ | $0.287_{0.080}$ |
| | 336 | $\underline{0.309}_{0.024}$ | $\mathbf{0.263}_{0.001}$ | $0.325_{0.065}$ |
| | 720 | $\underline{0.340}_{0.003}$ | $\mathbf{0.333}_{0.001}$ | $0.391_{0.060}$ |

Table 5: MSE (mean$_{std}$) for PatchTST with MissTSM under 70% masking.

| Dataset | Horizon Window | PatchTST + MissTSM | PatchTST + SAITS | PatchTST + Spline |
|---|---|---|---|---|
| ETTh2 | 96 | $\underline{0.322}_{0.004}$ | $0.548_{0.050}$ | $\mathbf{0.317}_{0.009}$ |
| | 192 | $\underline{0.382}_{0.011}$ | $0.561_{0.056}$ | $\mathbf{0.380}_{0.005}$ |
| | 336 | $\underline{0.384}_{0.008}$ | $0.468_{0.059}$ | $\mathbf{0.372}_{0.007}$ |
| | 720 | $0.621_{0.025}$ | $\underline{0.497}_{0.085}$ | $\mathbf{0.419}_{0.015}$ |
| ETTm2 | 96 | $\underline{0.213}_{0.006}$ | $0.405_{0.079}$ | $\mathbf{0.177}_{0.009}$ |
| | 192 | $\underline{0.266}_{0.003}$ | $0.447_{0.086}$ | $\mathbf{0.236}_{0.009}$ |
| | 336 | $\underline{0.315}_{0.004}$ | $0.494_{0.095}$ | $\mathbf{0.293}_{0.007}$ |
| | 720 | $\underline{0.432}_{0.025}$ | $0.529_{0.092}$ | $\mathbf{0.386}_{0.009}$ |
| Weather | 96 | $\underline{0.204}_{0.014}$ | $\mathbf{0.166}_{0.013}$ | $0.262_{0.122}$ |
| | 192 | $\underline{0.249}_{0.030}$ | $\mathbf{0.207}_{0.009}$ | $0.279_{0.092}$ |
| | 336 | $\underline{0.304}_{0.026}$ | $\mathbf{0.257}_{0.007}$ | $0.317_{0.075}$ |
| | 720 | $\underline{0.358}_{0.012}$ | $\mathbf{0.329}_{0.008}$ | $0.383_{0.071}$ |

### 5.2.3 Impact on Real-world Datasets

We observed that existing benchmark datasets used for forecasting represent a certain level of seasonality which makes it easier for imputation-based models to show adequate performance. However, in many

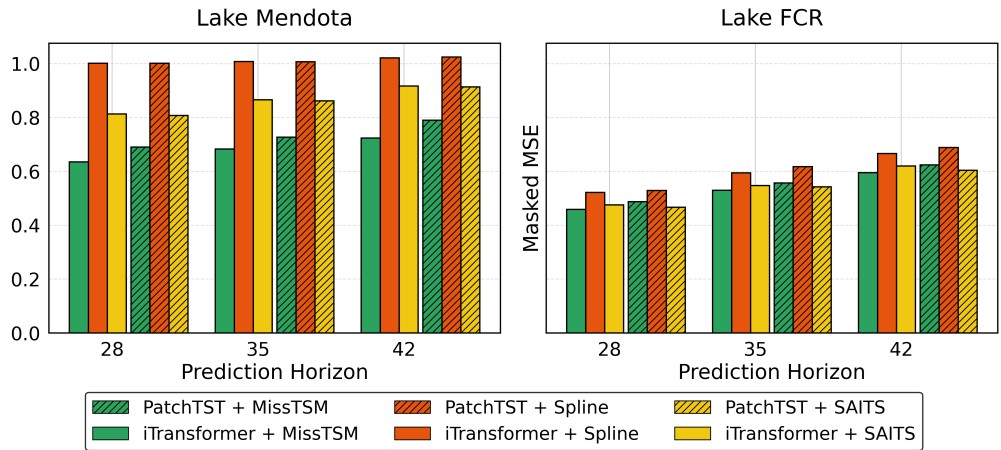

Figure 8: Forecasting performance comparison on Lake datasets across different prediction horizon windows. Lower MSE (y-axis) is better

real-world datasets such as those encountered in ecology, there are complex forms of temporal structure in the data beyond simple seasonality. We compare the performance of MissTSM integrated with two MTS models, iTransformer and PatchTST, on two Lake Datasets: Falling Creeks Reservoir and Mendota. Figure 8 reports masked MSE - MSE computed only on observed points - comparing MissTSM against imputation-based baselines. Competitive performance shown by MissTSM on both the real-world missing datasets further motivates the idea of imputation-free and model-agnostic approaches for IMTS modeling.

## 5.3 Ablations

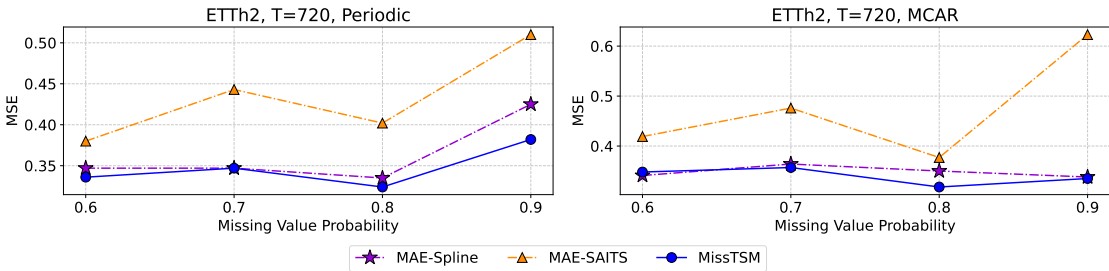

Figure 9: Ablations of MissTSM with and without TFI+MFAA layer on Forecasting datasets.

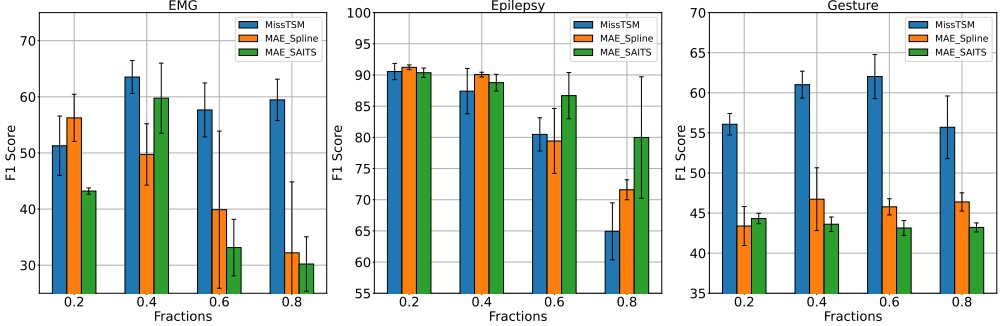

Figure 10: Ablations of MissTSM with and without the TFI+MFAA layer on the classification tasks.

In the ablation experiments, we evaluate the impact of integrating the MissTSM layer. We compare MAE with MissTSM against standard MAE (without MissTSM), using spline and SAITS imputation as additional baselines. The goal here is to understand the additional value of adding the MissTSM layer instead of modeling on imputed data. For forecasting (Fig. 9) and classification (Fig. 10), MissTSM consistently improves performance. In forecasting, MissTSM-MAE outperforms all MAE variants, while in classification, it is consistently comparable or superior across all three datasets.

## 6 Conclusion

We investigate the performance of existing IMTS models as well as our proposed MissTSM framework on a variety of datasets and tasks with varying conditions of missing values. We show that imputation-based frameworks built on simple imputations perform well when the amount of missingness is small or there is periodic structure in the data (e.g., in Weather data) that is easy to approximate. However, imputation-based approaches show poor performance at larger missing value fractions and when missing values have limited periodic patterns (e.g., on the lake datasets). We also show that MissTSM, which is an imputation-free and model-agnostic framework, demonstrates competitive performance across most datasets, tasks, and settings compared to imputation-based and existing imputation-free specialized models. We hope our findings could inspire further research into developing flexible, model-agnostic adapters for handling the challenges in irregularly-sampled time-series data.

**Limitations and Future Directions.** (1) A limitation of the MFAA layer is that it doesn't learn the non-linear temporal dynamics and relies on the subsequent transformer encoder blocks to learn the dynamics. Future work can explore modifications of the MFAA layer such that it can jointly learn the cross-channel correlations with the non-linear temporal dynamics. (2) Independent embedding of each time-feature token can become computationally expensive in high-dimensional multivariate systems.

### Broader Impact Statement

**Validation requirements and safety considerations when used in healthcare sectors** MissTSM is a decision-support tool and would require rigorous clinical validation to ensure reliable predictions. We would recommend that the model must be used in an "expert-in-the-loop" system

**Guidance on when the method is reliable vs when it may fail** The method is reliable in challenging scenarios with low periodicity, high missingness and high number of variates. On low variate, highly periodic data, a simpler approach like Spline + Backbone can be more effective

**The potential issue of over-trust in imputation-free outputs** We agree that this is a critical concern. Our recommendation would be to pair MissTSM's predictions with a reliable uncertainty quantification metric. This metric should reflect the sparsity (or quality) of the input data, warning the user when the predictions are of low confidence.

### Acknowledgments

This work was supported in part by NSF awards IIS-2239328 and DEB-2213550. This manuscript has been authored by UT-Battelle, LLC, under contract DE-AC05-00OR22725 with the US Department of Energy (DOE). The US government retains and the publisher, by accepting the article for publication, acknowledges that the US government retains a nonexclusive, paid-up, irrevocable, worldwide license to publish or reproduce the published form of this manuscript, or allow others to do so, for US government purposes. DOE will provide public access to these results of federally sponsored research in accordance with the DOE Public Access Plan (https://www.energy.gov/doe-public-access-plan).

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

# A    Dataset Details

## A.1    Forecasting

**ETT.** The ETT (Zhou et al., 2021) dataset captures load and oil temperature data from electricity transformers. ETTh2 includes 17,420 hourly observations, while ETTm2 comprises 69,680 15-minute observations. Both datasets span two years and contain 7 variates each.

**Weather.** Weather (Weather, 2021) is a 10-minute frequency time-series dataset recorded throughout the year 2020 and consists of 21 meteorological indicators, like humidity, air temperature, etc.

**Solar-Energy.** Solar-energy (Lai et al., 2018) dataset captures solar power generation data from 137 photovoltaic (PV) plants in the year 2006, with data sampled at a resolution of every 10 min, providing rich MTS data. We report results on Solar-Energy dataset in the appendix only, as supplementary forecasting evaluation.

Following Wu et al. (2021) and Lai et al. (2018), we use a train-validation-test split of 6:2:2 for the ETT and Solar-Energy datasets and 7:1:2 for the Weather dataset.

**Ecology Dataset - Falling Creek Reservoir (FCR).** The FCR dataset is generated by combining data from the following data packages, Carey et al. (2023a; 2024); Carey & Breef-Pilz (2023b;a); Carey et al. (2023b), published in the Environmental Data Initiative repository. The dataset contains daily median meteorological and water quality observations from FCR (Virginia, USA), collected over a period of 3 years spanning from 2018-08-01 to 2021-12-22. As part of the pre-processing, we removed variables with more than 90% missing values. Specifically, two columns were excluded with the remaining dataset consisting of 1240 time points with an overall missing rate of 53.12%.

**Ecology Dataset - Mendota.** The Mendota dataset contains daily averaged meteorological and water quality observations from Lake Mendota (Dane County, WI, USA). Data were collected using an instrumented buoy at the surface of the water during the ice-free season from July 2006 through November 2023 (Magnuson et al., 2024). The data were cleaned by filtering out flagged values before use. We applied preprocessing and removed variables with more than 90% missing data. This resulted in dropping three columns with 99% missing values, and one column with 98.1% missing values. The resulting dataset contains 6321 time points, with an overall missing rate of 50.07%.

For both datasets, we used a split of 70% for training, 10% for validation, and 20% for testing.

## A.2    Classification

**Epilepsy.** Epilepsy (Andrzejak et al., 2001) contains univariate brainwaves (single-channel EEG) sampled from 500 subjects (with 11,500 samples in total), with each sample classified as having epilepsy or not (binary classification).

**Gesture.** Gesture (Liu et al., 2009) dataset consists of 560 samples, each having 3 variates (corresponding to the accelerometer data) and each sample corresponding to one of the 8 hand gestures (or classes)

**EMG.** EMG (Goldberger et al., 2000a) dataset contains 163 EMG (Electromyography) samples corresponding to 3-classes of muscular diseases.

We make use of the following readily available data splits (train, validation, test) for each of the datasets: **Epilepsy** = 60 (30 samples per each class)/20 (10 samples per each class)/11420 (Train/Val/Test) **Gesture** = 320/20/120 (Train/Val/Test) **EMG** = 122/41/41 (Train/Val/Test)

**Physio-Net Dataset:** PhysioNet-2012 Mortality Prediction Challenge (Silva et al., 2012) contains 12k multivariate clinical time-series samples that were collected from patients in ICU. The time-series contains 37 variables, such as temperature, heart rate, blood pressure, etc. that can vary depending on the type of patient. Each of the samples are recorded during the first 48 hours of admission in ICU. PhysioNet has a high degree of 80% missing values. We follow the experimental setup in Du et al. (2023), and split the dataset into 80%, 10% and 10% train/val/test split.

**P12 Dataset** Derived from the PhysioNet 2012 Challenge, the P12 (Goldberger et al., 2000b) dataset includes multivariate ICU time-series data for 11,988 patients after filtering out 12 invalid entries. Each patient record contains up to 48 hours of sensor data across 36 physiological variables (excluding weight), plus a 9-dimensional static vector with demographic attributes such as age and gender. The binary label indicates ICU length of stay: $\leq 3$ days (negative) vs. $\geq 3$ days (positive).

**P19 Dataset** (Reyna et al., 2020) This dataset is part of the PhysioNet 2019 Sepsis Early Prediction Challenge and contains longitudinal EHR data from 38,803 ICU patients. Each patient record comprises multivariate time-series data with 34 clinical variables, sampled at irregular intervals, alongside a static vector containing demographic and admission information (e.g., age, gender, ICU type, hospital-to-ICU delay, ICU length of stay). A binary label indicates whether the patient will develop sepsis within the next 6 hours.

### A.3 Synthetic Masked Data Generation

**Random Masking**: We generated masks by randomly selecting data points across all variates and time-steps, assigning them as missing with a likelihood determined by p (masking fraction). The selected data points were then removed, effectively simulating missing values at random. For multiple runs, we created multiple such versions of the synthetic datasets and compared all baseline methods and MissTSM on the same datasets.

**Periodic Masking**: We use a sine curve to generate the masking periodicity with given phase and frequency values for different features. Specifically, the time-dependent periodic probability of seeing missing values is defined as $\hat{p}(t) = p + \alpha(1-p)\sin(2\pi\nu t + \phi)$, where, $\phi$ and $\nu$ are randomly chosen across the feature space, $\alpha$ is a scale factor, and p is an offset term. We vary p from low to high values to get different fractions of periodic missing values in the data. To implement this masking strategy, each feature in the dataset was assigned a unique frequency, randomly selected from the range [0.2, 0.8]. This was done to reduce bias and increase randomness in periodicity across the feature space. Additionally, the phase shift was chosen randomly from the range $[0, 2\pi]$. This was applied to each feature to offset the sinusoidal function over time. Like frequency, the phase value was different for different features. This generated a periodic pattern for the likelihood of missing data.

## B Baselines

**Regular MTS Methods.** iTransformer (Liu et al., 2023), PatchTST (Nie et al., 2022b), DLinear (Zeng et al., 2023), Autoformer (Wu et al., 2021), SimMTM (Dong et al., 2024), MAE (He et al., 2022) (adapted for Time series similar to Ti-MAE (Li et al., 2023)). We use the default parameters reported for each models in their official code for the respective datasets. We use the parameters reported in Table 6 and Table 8 for the MAE implementation, as code for Ti-MAE (Li et al., 2023) is not publicly available.

**Imputation Methods** $2^{nd}$-order spline imputation (McKinley & Levine, 1998), k-Nearest Neighbor (kNN) (Tan et al., 2019) (with neighbors=10), SAITS (Du et al., 2023), and BRITS (Cao et al., 2018). For SAITS, we use the following setting in our experiments `n_layers=2`, `embedding_dim=256`, `num_heads=4`. For BRITS, we use `hidden_dimension=64`. We follow the official code for implementation of these models.

**IMTS Methods** GRU-D (Che et al., 2018), Latent ODE (Rubanova et al., 2019). We used the available GRU-D re-implementation as the official code was not available. For Latent ODE we used their official code. The other baselines used are the following, SeFT (Horn et al., 2020), mTAND (Shukla & Marlin, 2021), Raindrop (Zhang et al., 2021), MTGNN (Wu et al., 2020)

## C Implementation Details

The experiments have been implemented in PyTorch using NVIDIA TITAN 24 GB GPU. The baselines have been implemented following their official code and configurations. In the default implementation code of MissTSM with MAE, we integrate the MissTSM layer directly within the backbone without the linear projection to the original input shape. We consider Mean Squared Error (MSE) as the metric for time-series forecasting and F1-score, AUROC, AUPRC for the classification tasks. We generate five different versions of

synthetic data and report the average metric from these 5 runs along with their standard deviation, wherever applicable.

## C.1 Hyper-parameter Details

The hyperparameters for MissTSM are selected after integration with the base model, with embedding dimensions searched in the range [4, 128] and the number of attention heads in the range [1, 8]. For the base models, we use the default configurations for the iTransformer (Liu et al., 2023) and PatchTST (Nie et al., 2022b) models. For the MAE base model, we start with the same set of hyper-parameters as reported in the SimMTM paper (see Table 6) as initialization, and then search for the best learning rate in factors of 10, encoder/decoder layers in the range [2, 4], number of heads in the range [2,16] and embedding dimensions in the range [4, 128]. Note that we only perform hyper-parameter tuning on 100% (or fully observed) data, and use the same hyper-parameters for all experiments involving the dataset, such as different missing value probabilities. Table 7 captures the MissTSM parameters across datasets. Note that, for classification datasets (EMG, Epilepsy, Gesture), we use the same set of parameters with $q_{\dim} = v_{\dim} = k_{\dim} = 32$ and $n_{\mathrm{head}} = 16$. Table 8 presents the sensitivity analysis of the MissTSM-MAE model with respect to key hyperparameters (measured using MSE).

Table 6: Hyperparameters for Forecasting and Classification Tasks used for base MAE model

| Hyperparameter | ETTh2 | ETTm2 | Weather | Classification |
|---|---|---|---|---|
| # Encoder Layers | 2 | 3 | 2 | 3 |
| # Decoder Layers | 2 | 2 | 2 | 2 |
| # Encoder Heads | 8 | 8 | 8 | 16 |
| # Decoder Heads | 4 | 4 | 4 | 16 |
| Encoder Embed Dim | 8 | 8 | 64 | 32 |
| Decoder Embed Dim | 32 | 32 | 32 | 32 |

Table 7: Hyperparameters of the MissTSM layer

| Base Model | $q_{\mathbf{dim}}$ | $k_{\mathbf{dim}}$ | $v_{\mathbf{dim}}$ | $n_{\mathbf{heads}}$ | Dataset |
|---|---|---|---|---|---|
| PatchTST | 128 | 8 | 8 | 1 | |
| iTransformer | 128 | 8 | 8 | 1 | ETTh2 & ETTm2 |
| MAE | 8 | 8 | 8 | 8 | |
| PatchTST | 128 | 8 | 8 | 1 | |
| iTransformer | 128 | 8 | 8 | 1 | Weather |
| MAE | 64 | 32 | 32 | 8 | |
| MAE | 32 | 32 | 32 | 16 | EMG, Epilepsy, Gesture |
| MAE | 64 | 64 | 64 | 2 | PhysioNet, P12, P19 |
| PatchTST | 16 | 8 | 8 | 1 | |
| iTransformer | 128 | 8 | 8 | 1 | FCR & Mendota |

## C.2 Forecasting experiments

The models were trained with the MSE loss, using the Adam (Kingma, 2014) optimizer with a learning rate of 1e-3 during pre-training for 50 epochs and a learning r ate of 1e-4 during fine-tuning with an early stopping counter of 3 epochs. Batch size was set to 16. All the reported missing data experiment results are obtained over 5 trials (5 different masked versions). During fine-tuning for different Prediction lengths (96, 192, 336, 720), we used the same pre-trained encoder and added a linear layer at the top of the encoder.

For our experiments, we use the Latent ODE model with the following configuration. The number of training iterations is set to 50, using a dataset size of 10,000 samples and a batch size of 32. The latent state dimension

Table 8: Hyper-parameter sensitivity of MAE model integrated with MissTSM on ETTh2 with 70% Masking Fraction, MCAR. Best results shown in bold, second best underlined. Hyper-parameter settings used in the paper are italicized.

|  | Enc. Heads | | | Enc. Layers | | | Enc. Embed Dim | | |
|---|---|---|---|---|---|---|---|---|---|
|  | 1 | 4 | 8 | 1 | 2 | 3 | 8 | 16 | 32 |
| 96 | 0.246 | **0.245** | *0.246* | 0.249 | ***0.243*** | 0.244 | ***0.243*** | 0.248 | 0.285 |
| 192 | **0.261** | 0.273 | *0.266* | 0.287 | ***0.267*** | 0.271 | *0.267* | **0.266** | 0.340 |
| 336 | 0.312 | **0.279** | *0.310* | **0.294** | *0.392* | 0.307 | *0.392* | **0.316** | 0.369 |
| 720 | **0.326** | 0.346 | *0.333* | 0.351 | ***0.323*** | 0.355 | ***0.323*** | 0.338 | 0.446 |

|  | Dec. Heads | | | Dec. Layers | | | Dec. Embed Dim | | |
|---|---|---|---|---|---|---|---|---|---|
|  | 1 | 4 | 8 | 1 | 2 | 3 | 8 | 16 | 32 |
| 96 | 0.261 | ***0.243*** | 0.252 | 0.276 | ***0.242*** | 0.248 | 0.250 | 0.259 | ***0.243*** |
| 192 | 0.276 | ***0.267*** | 0.272 | **0.266** | *0.268* | 0.268 | **0.257** | 0.272 | *0.267* |
| 336 | 0.319 | *0.392* | **0.301** | **0.262** | *0.352* | 0.271 | 0.289 | **0.266** | *0.392* |
| 720 | 0.324 | ***0.323*** | 0.330 | **0.323** | *0.364* | 0.341 | 0.353 | 0.384 | ***0.323*** |

is 16, and the dataset used is ETTh2. We operate in forecasting mode, using a sequence length of 336 and a prediction horizon of 96. The recognition model (ODE/RNN) has 30 dimensions and 3 layers, while the generative ODE function has 3 layers. Each ODE function layer has 300 units, and the GRU update networks have 100 units.

For the lake dataset, both the iTransformer and PatchTST models were trained using a sequence length of 21 and prediction lengths of 28, 35, and 42. The iTransformer model was trained with a batch size of 16 and a learning rate of 0.0001. The model uses 2 encoder layers, 8 attention heads, and a model dimension of 128. The PatchTST model was trained with the same batch size and learning rate. Its encoder consisted of 3 layers, 4 attention heads, and a model dimension of 16, with a dropout rate of 0.3. The patch length was set to 16, with a stride of 8.

### C.3    Classification experiments

The models were trained using the Adam (Kingma, 2014) optimizer, with MSE as the loss function during pre-training and Cross-Entropy loss during fine-tuning. During fine-tuning, we plugged a 64-D linear layer at the top of the pre-trained encoder. We pre-trained and fine-tuned for 100 epochs. For the GRU-D experiments, we use the available re-implementation of the code, with hidden size as 16. For the PhysioNet experiments, we follow the same evaluation setup as proposed in Du et al. (2023). The baselines imputation models use a simple RNN model as a classification model on top of the imputed data.

### C.4    Discussion on Design Choices

We discuss some of the design choices behind the MissTSM layer.

**Why cross-attention instead of a self-attention at each timestep?** In our current formulation, we perform a masked cross-attention at every timestep using a single learnable query. While we agree that the conventional approach of applying self-attention can capture better cross-variate interactions, we specifically decided to learn a single query vector to reduce the computational cost of learning attention weights (we only need $N \times 1$ attention weights, instead of a full $N \times N$ matrix). While we empirically show that our simple attention architecture still captures valuable information across variates, our work can be easily extended in future works to model all variate-pairs interactions, e.g., by introducing a hierarchical or grouped aggregation mechanism that first models variate dependencies for a single time-step and then performs a global pooling across time-steps.

**Why a single time-independent query?** The goal of the MFAA layer is to learn a feature representation at each time-step using the observed variates while ignoring the missing variates. It does so by using a learnable query. The goal of learning this query vector can be equated to the task of extracting the best

representation from the observed features at every timestep. Since this task can be assumed to be invariant of time (i.e., the same feature extractor can be used to learn representations of observed variates at any time-step), we consider learning a single time-independent query. Alternatively, if we used time-dependent queries, it would require learning $T$ separate query vectors, which would increase the number of learnable parameters. Moreover, by learning a different query for every time-step, the model would be forced to learn $T$ different rules, potentially making it susceptible to overfitting to artifacts at specific time indices.

## D   Complexity and Scalability Analysis

### D.1   Theoretical Analysis of Computational Costs

We analyze the computational and memory complexity of the proposed MissTSM module. Let $L$ represent the context length, $N$ the number of variates (features), $d$, the embedding dimension, $h$, the number of heads, and $Q$, $K$, $V$ are the Query, Key and Value matrices

**Time Complexity** The initial feature embedding (i.e. the Time-feature independent embedding TFI) processes each of the $N$ variates at each of the $L$ time steps, mapping them to a $d$-dimensional space. This operation has a time complexity of $O(LNd)$. The MFAA layer performs cross-attention across variates at each time-step, where a single query attends to $N$ embedded variates. The complexity is dominated by the linear projections to $Q$, and $K$ or $V$, $O(d^2)$ and $O(Nd^2)$ respectively. It results in a complexity of $O(LNd^2)$. The final projection layer maps the $d$-dimensional output to the variate embedding space, adding $O(LdN)$ to the time complexity, overall resulting in a total time complexity of MissTSM as $O(L(Nd^2 + dN)), = O(LNd^2)$

Table 9: Time, memory, and parameter complexity of MissTSM. Here, $L$ is the context length, $N$ the number of variates, $h$, number of heads and $d$ the embedding dimension. $d_n$ and $d_l$ are hidden and latent dimensions of mTAND, respectively. $G$ and $l_{inner}$ corresponds to the number of groups within the DMSA blocks and number of layers within each block in SAITS

| Model | Time | Memory | Params |
|-------|------|--------|--------|
| MissTSM | $\mathcal{O}(LNd^2)$ | $\mathcal{O}(LNd)$ | $\mathcal{O}(d^2 + Nd)$ |
| mTAND | $\mathcal{O}(L^2(d^2 + h(N + d_n)) + Ld_l d_n)$ | $\mathcal{O}(L(N + d_n + d_l + d) + hL^2)$ | $\mathcal{O}(d^2 + hNd_n + hd_n^2 + d_n d_l)$ |
| SimMTM | $\mathcal{O}(NLdE(L + d))$ | $\mathcal{O}(Ld + hL^2)$ | $\mathcal{O}(Ed^2 + L^2 d^2)$ |
| SAITS | $\mathcal{O}(L(Nd + N^2 + L) + Gl_{inner}L^2d)$ | $\mathcal{O}(L(N + d) + Gl_{inner}(Ld + hL^2))$ | $\mathcal{O}(l_{inner}d^2 + N(d + N + L))$ |

**Parameter and Memory Complexity** The main learnable parameters lie in the cross-attention layer (approximately $4d^2$ parameters from the $Q$, $K$, $V$ and the output projection layers), the $TFI$ embedding ($2Nd$ parameters from the weight and bias matrices), and the final linear layer ($dN$ parameters). Thus, the total parameter count is $d + 4d^2 + 2Nd + 2d$, yielding an overall parameter complexity of $O(d^2 + Nd)$ - quadratic in $d$ and linear in $N$

The largest intermediate tensors during the forward pass are the embedded features ($O(LNd)$) and the $Q,K,V$ tensors ($O(LNd)$), and the attention scores ($O(LN)$). Hence, the peak activation memory scales as $O(LNd)$, which is linear in $N$.

Table 9 summarizes the time, memory and parameter complexity of MissTSM, compared against three baselines - mTAND (IMTS method), SimMTM (MTS method) and SAITS (imputation method). The scalability of MissTSM is primarily governed by three factors - embedding dimension, number of variates and sequence length. In contrast, the baselines exhibit complexity that depends on multiple interacting components within each dimension (time, space, parameters), making their scalability less straightforward.

### D.2   Empirical Analysis of Computational Costs

To evaluate the efficiency of different IMTS pipelines, we benchmark their training runtime, inference throughput, and peak GPU memory usage. Table 10 presents these metrics across models.

Table 10: Computational cost comparison between different pipelines and ours for IMTS modeling. We consider a classification task on the PhysioNet P19 dataset

| Pipeline | Training Time (avg. epoch, sec) ↓ | Inference Throughput (samples/sec) ↑ | Peak Active Tensor Memory (train, GB) ↓ | # Learnable Params | AUROC |
|---|---|---|---|---|---|
| **SAITS$_{+MAE}$** | $13.277_{0.578} + 0.279_{0.163}$ $= 13.556_{0.601}$ | $438.135_{47.731}$ | **0.74** | $1,373,576 + 363,236$ $= 1,736,812$ | 77.40 |
| **mTAND** **Raindrop** | $5.95_{0.117}$ $17.651_{0.554}$ | $9593.36_{121.44}$ $964.75_{88.91}$ | 1.6 1.4 | 142,370 1,947,804 | 82.9 87.6 |
| **MissTSM$_{+MAE}$** | **$0.322_{0.077}$** | **$19371.58_{1315.92}$** | 1.866 | **365,506** | **88.8** |

The training time is reported as the average per epoch over five trials. Inference throughput is computed as the total number of test samples divided by the total inference time. For two-stage IMTS pipeline, the total inference time is the sum across both the stages. We report the peak GPU memory allocated using PyTorch. For two-stage pipeline, we report the maximum GPU memory usage across both stages. All the experiments were conducted on Ubuntu 18.04, Python 3.9, PyTorch 1.12 (CUDA 12.8) and a single NVIDIA TITAN 24GB GPU, using a constant batch size of 64 for all models and with mixed precision disabled

The empirical results demonstrate that the MissTSM+MAE framework is approximately $42\times$ faster per training epoch and achieves over $44\times$ higher inference throughput than the two-stage SAITS+MAE pipeline. However, the peak memory of the MissTSM+MAE (1.866 GB) is slightly higher than the SAITS+MAE pipeline (0.74 GB), which can be further improved using more efficient code optimization in future works.

### D.3  MissTSM Scalability

We conducted an empirical study of runtime and memory usage as $N$ (number of variates) increases, to analyze the scalability of our $T \times N$ tokenization in the $TFI$ layer. We generated multiple synthetic datasets of approximately 56k observations with a 70% MCAR missing fraction, varying $N$ in $10, 20, 50, 100, 200, 500, 800, 1000$. For this analysis, we considered MissTSM integrated with an iTransformer backbone and compare its performance with a baseline approach of using SAITS imputation with iTransformer. We report the peak GPU memory and average time per epoch (Figure 12) as well as per forward pass (Figure 11). All the results are summarized in Table 11. In Figure 12, we compare the per epoch metrics between MissTSM+iTransformer and iTransformer model, and in Figure 11, we compare the metrics per forward pass between iTransformer and the MissTSM layer. The sequence length $T$ and the embedding dimension $d$ were held constant to 336 and 16 respectively, to isolate the effect of $N$. Note that the reported baseline results reflect only the runtime and memory requirements of the iTransformer model itself, excluding the additional computational cost associated with the SAITS imputation process, for ease of analysis.

Table 11: Scalability analysis with respect to $N$.

| N | Forward Pass (per epoch) | | | | Forward + Backward Pass (per epoch) | | | |
|---|---|---|---|---|---|---|---|---|
| | Time (s) | | Peak GPU (GB) | | Time (s) | | Peak GPU (GB) | |
| | MissTSM | iTrans | MissTSM | iTrans | MissTSM+iTrans | iTrans | MissTSM+iTrans | iTrans |
| 10 | 2.095 | 1.784 | 0.080 | 0.064 | 22.416 | 13.495 | 0.271 | 0.177 |
| 20 | 2.954 | 1.911 | 0.149 | 0.114 | 27.312 | 16.275 | 0.410 | 0.236 |
| 50 | 6.004 | 3.163 | 0.359 | 0.275 | 49.903 | 18.514 | 0.847 | 0.427 |
| 100 | 10.227 | 12.487 | 0.711 | 0.576 | 96.618 | 48.387 | 2.764 | 1.925 |
| 200 | 20.140 | 24.188 | 1.410 | 1.309 | 174.575 | 69.040 | 3.313 | 1.636 |
| 500 | 52.917 | 62.014 | 3.515 | 4.536 | 456.296 | 185.875 | 9.182 | 5.034 |
| 800 | 83.468 | 113.758 | 5.613 | 9.309 | 1204.278 | 335.082 | 17.028 | 10.403 |

From Figure 11, we observe that during the forward pass, the MissTSM layer scales almost linearly with $N$, consistent with the theoretical complexity in Table 9. However, Figure 12 shows that the runtime and

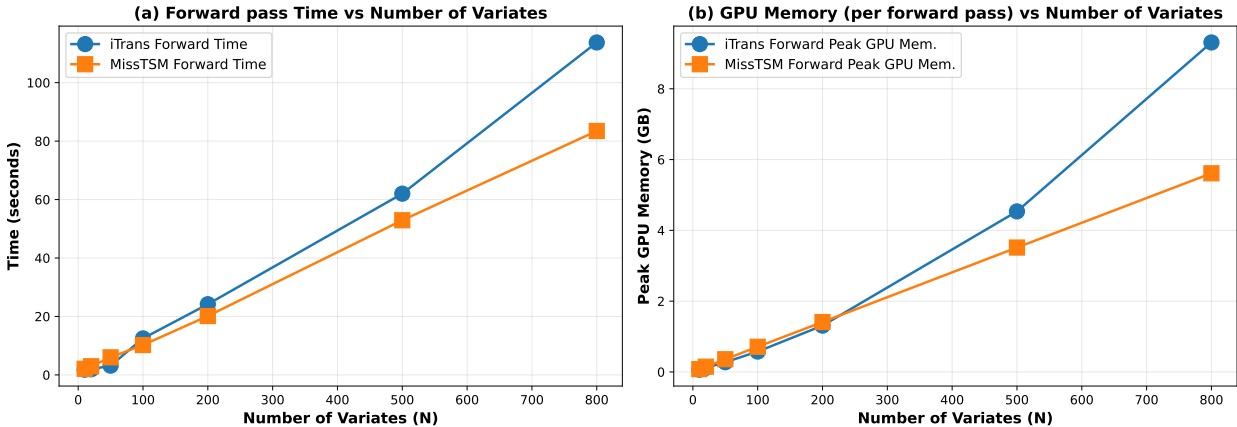

Figure 11: Time and peak GPU memory scaling w.r.t. $N$, observed per forward pass

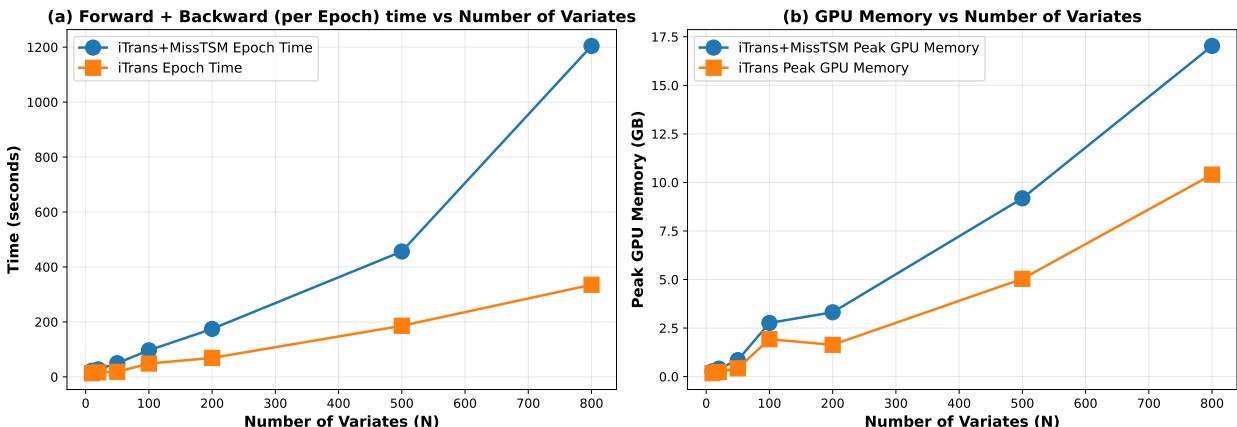

Figure 12: Time and peak GPU memory scaling w.r.t. $N$, observed per epoch

memory for the combined forward and backward pass scale linearly only up to a certain point. Till $N = 100$, iTransformer and MissTSM+iTransformer models scale almost in parallel, but diverges after that. Comparing the scaling curves in Figures 11 and 12, and the forward versus forward+backward metrics for iTransformer, we find that the backward pass dominates both runtime and peak GPU memory, leading to a sharp increase in runtime after $N > 100$. On a single 24 GB NVIDIA RTX, training MissTSM becomes limiting at $N \approx 1000$. However, note that $N > 100$ represents an extreme setup involving very large number of variates, while for common practical ranges of $N$ (e.g., $N < 50$), the computational overhead of backpropagation with MissTSM remains minimal.

### D.4 Imputation Time and Error Propagation - Case Study

We conduct a case study to examine the trade-offs between models for IMTS modeling in terms of computational complexity and efficiency, and to investigate potential correlations between imputation errors and downstream performance. We consider a classification task on the Epilepsy dataset that is 80% masked under MCAR. Spline and SAITS are the imputation techniques and SimMTM is the MTS model used. We report the total modeling time as the sum of imputation time and the time-series model training time. The experiments are conducted on NVIDIA TITAN 24 GB GPUs.

In Table 12, we observe that, while SimMTM integrated with SAITS achieves a higher F1 score than Spline, but the total imputation time for SAITS is significantly higher than that of Spline. This additional computational overhead substantially increases the overall modeling time. Moreover, SAITS has approximately 1.3 million

Table 12: Total computational cost comparison between MissTSM-MAE (model-agnostic and imputation-free) and SimMTM (MTS model) for an IMTS classification task

| Model | Imp. Model | Imp. Time (sec) | MTS Model Train Time (sec) | Total Time (sec) | F1 Score |
|---|---|---|---|---|---|
| **SimMTM** | SAITS | $949 \pm 42.9$ | $397.59 \pm 2.64$ | $1346.59 \pm 45.54$ | $\underline{61.0 \pm 9.20}$ |
| | Spline | $8.74 \pm 0.38$ | $397.59 \pm 2.64$ | $\underline{406.33 \pm 3.02}$ | $59.16 \pm 3.67$ |
| **MissTSM** | N/A | N/A | $346.8 \pm 7.32$ | $\mathbf{346.8 \pm 7.32}$ | $\mathbf{64.93 \pm 4.57}$ |

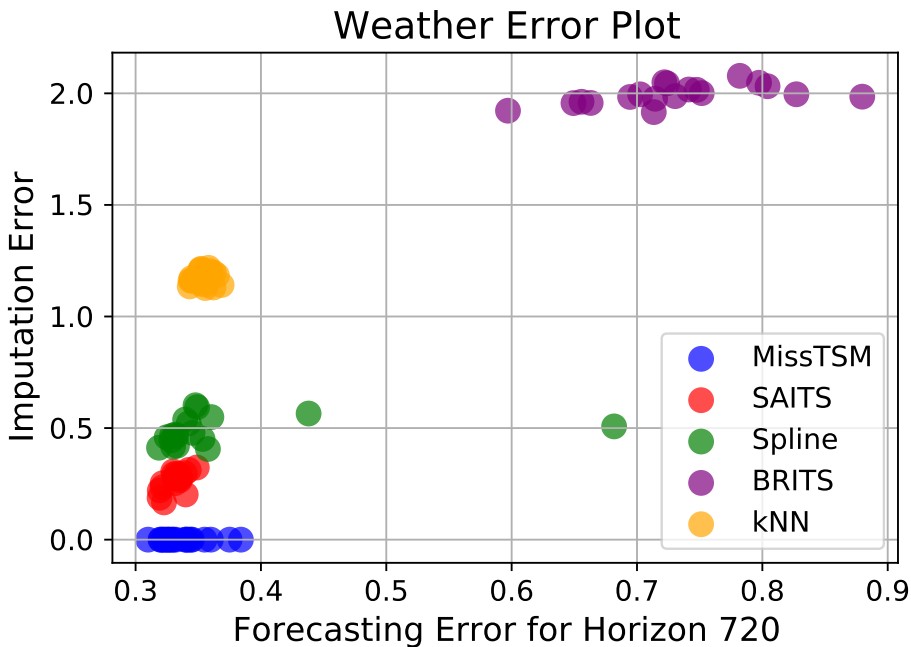

Figure 13: Imputation error vs Forecasting error across 5 trials for 4 missing fractions, 0.6, 0.7, 0.8, 0.9

trainable parameters, further increasing the overall model complexity of the time-series modeling task. This highlights a potential trade-off between imputation efficiency and model complexity. With our proposed model-agnostic approach, we do not have the extra overhead of imputation complexity. Interestingly, the MissTSM-integrated model leads to superior performance.

Figure 13 captures the propagation of imputation errors and forecasting errors for Weather dataset at 720 forecast horizon. It indicates that there is an overall positive correlation between the imputation error and forecasting errors, thereby demonstrating propagation of the imputation errors into the downstream time-series models.

# E Additional Results

## E.1 Embedding of 1D data and the Effect of Varying Embedding Sizes

To understand the usefulness of mapping 1D data to multi-dimensional data in TFI embedding, we present (in Table 13) an ablation comparing performances on ETTh2 with and without using high-dimensional projections in TFI Embedding under the no missing value scenario. Projecting 1D scalars independently to higher-dimensional vectors may look wasteful at the time of initialization of TFI Embedding, when the context of time and variates are not incorporated. However, it is during the cross-attention stage (using MFAA layer or later using the Transformer encoder block) that we can leverage the high-dimensional embeddings to store richer representations bringing in the context of time and variate in which every data point resides.

From Table 13, we can see that TFI embedding with 8-dimensional vectors consistently outperform the ablation with 1D representations, empirically demonstrating the importance of high-dimensional projections in our proposed framework.

Table 13: Effect of TFI Embedding with embedding size=1 and embedding size=8 under no masking scenario. Dataset=ETTh2

| Horizon | TFI Embedding with embedding size = 1 | TFI Embedding with embedding size = 8 |
|---------|---------------------------------------|---------------------------------------|
| 96  | $0.283_{\ 0.048}$ | $\mathbf{0.245}_{\ 0.011}$ |
| 192 | $0.285_{\ 0.078}$ | $\mathbf{0.260}_{\ 0.023}$ |
| 336 | $0.319_{\ 0.023}$ | $\mathbf{0.300}_{\ 0.016}$ |
| 720 | $0.378_{\ 0.022}$ | $\mathbf{0.334}_{\ 0.032}$ |

### E.2 Effectiveness of TFI Embedding

To understand the standalone performance of the MFAA layer, we conducted a simple ablation study where we fixed the weights of the TFI embedding layers to a constant value of 1 with a bias of 0, to simulate a scenario of no TFI embedding. This corresponds to using every time-feature combination as a scalar value rather than a multidimensional embedding. For this study, we considered the ETTh2 dataset across 60-90% MCAR missing fractions. The results are summarized in Table 14. We can observe that MissTSM suffers

Table 14: Performance comparison with and without TFI embedding across fractions and horizons

| Mask Ratio | Horizon | W/o TFI | With TFI |
|------------|---------|---------|----------|
| 60% | 96  | 0.53 | **0.24** |
|     | 192 | 0.47 | **0.26** |
|     | 336 | 0.49 | **0.28** |
|     | 720 | 0.83 | **0.33** |
| 70% | 96  | 0.42 | **0.25** |
|     | 192 | 0.46 | **0.27** |
|     | 336 | 0.58 | **0.30** |
|     | 720 | 1.73 | **0.35** |
| 80% | 96  | 0.47 | **0.26** |
|     | 192 | 0.68 | **0.28** |
|     | 336 | 1.02 | **0.29** |
|     | 720 | 0.69 | **0.37** |
| 90% | 96  | 0.50 | **0.32** |
|     | 192 | 0.68 | **0.35** |
|     | 336 | 1.05 | **0.32** |
|     | 720 | 0.66 | **0.35** |

significantly on removing the TFI embeddings, suggesting the importance of using learnable multidimensional embeddings for every time-feature combination rather than scalar values.

### E.3 Additional Forecasting Dataset

To further strengthen our understanding of the behaviors of regular IMTS models under the two-step pipeline, and to explore the potential of model-agnostic imputation-free approaches, we conduct forecasting experiments on another dataset, Solar-Energy (Lai et al., 2018), using the MCAR masking scheme over two missing data

fractions - 60 and 70%. Figure 14 summarizes the results. As expected, the performance of traditional MTS methods degrades with increasing missingness, reflecting their reliance on separate imputation strategies. While IMTS methods provide a practical solution for partially observed time series, they tend to underperform compared to regular MTS models in long-range forecasting scenarios. This is where we believe the gap could potentially be addressed by imputation-free, model-agnostic approaches such as MissTSM.

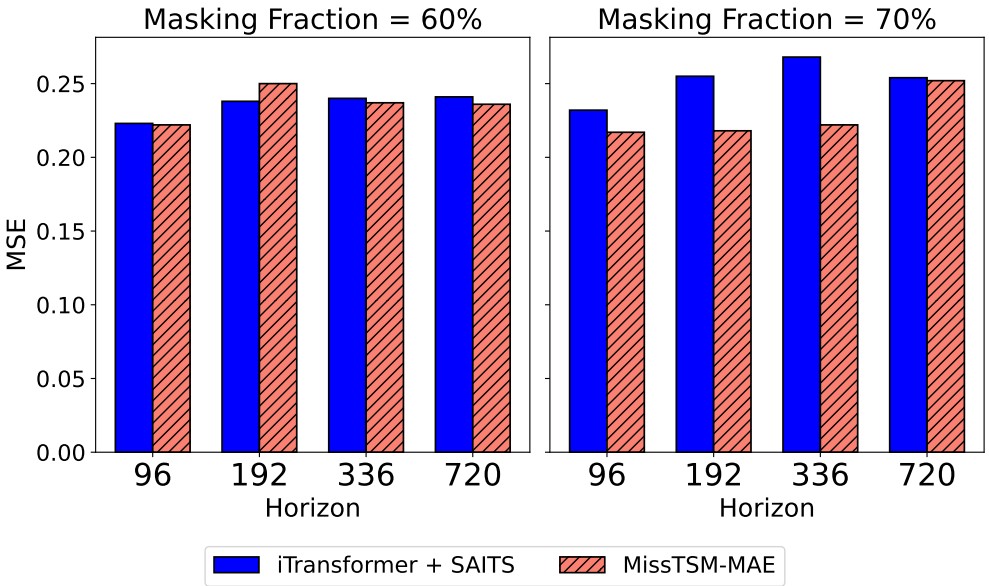

Figure 14: Results on the Solar-Energy dataset with MCAR masking across four horizons at 60% and 70% masking fractions. Lower MSE is better

### E.4   Analyzing the Impact of Forecast Horizon

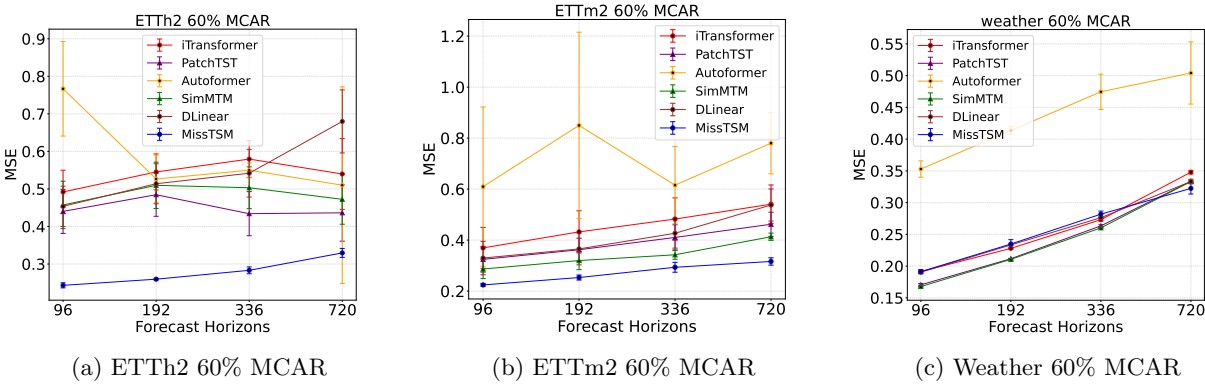

(a) ETTh2 60% MCAR                (b) ETTm2 60% MCAR                (c) Weather 60% MCAR

Figure 15: Forecasting performance with horizon length $T \in \{96, 192, 336, 720\}$ and fixed lookback length $S = 336$. Baseline models are imputed with SAITS.

To analyze the performance of regular MTS models under IMTS setting with increasing forecasting horizon $T = \{96, 192, 336, 720\}$, we consider a 60% MCAR masking scheme with five regular MTS models and MissTSM integrated-MAE, as shown in Figure 15. As expected, a common trend across all three datasets is that the forecasting MSE increases with the forecasting horizon for all methods. While this pattern generally holds for most MTS models, a few exhibit irregular behavior, with noticeably higher variability in their predictions. This could suggest sensitivity to the quality of the underlying imputation model or a need for greater robustness in certain MTS approaches.

Table 15: Performance (MAE) of iTransformer with different imputation methods (LOCF, Spline, SAITS) compared to MissTSM on the ETTm2 dataset across varying missing fractions and forecasting horizons.

| | Mask Ratio | iT + LOCF | iT + Spline | iT + SAITS | MissTSM |
|---|---|---|---|---|---|
| 96 | 60% | $\mathbf{0.18}_{0.002}$ | $\mathbf{0.18}_{0.002}$ | $0.37_{0.08}$ | $\underline{0.22}_{0.005}$ |
| | 70% | $\mathbf{0.18}_{0.002}$ | $\mathbf{0.18}_{0.006}$ | $0.46_{0.11}$ | $\underline{0.23}_{0.006}$ |
| | 80% | $\underline{0.19}_{0.002}$ | $\mathbf{0.18}_{0.008}$ | $0.64_{0.14}$ | $0.23_{0.012}$ |
| | 90% | $\underline{0.21}_{0.002}$ | $\mathbf{0.20}_{0.009}$ | $0.86_{0.22}$ | $0.24_{0.034}$ |
| 192 | 60% | $\underline{0.25}_{0.003}$ | $\mathbf{0.24}_{0.005}$ | $0.43_{0.08}$ | $\underline{0.25}_{0.009}$ |
| | 70% | $\underline{0.26}_{0.002}$ | $\mathbf{0.25}_{0.007}$ | $0.54_{0.14}$ | $\underline{0.26}_{0.018}$ |
| | 80% | $\mathbf{0.25}_{0.002}$ | $\mathbf{0.25}_{0.005}$ | $0.69_{0.13}$ | $\underline{0.27}_{0.013}$ |
| | 90% | $\mathbf{0.27}_{0.003}$ | $\mathbf{0.27}_{0.01}$ | $\underline{0.93}_{0.19}$ | $\mathbf{0.27}_{0.010}$ |
| 336 | 60% | $\underline{0.30}_{0.006}$ | $\mathbf{0.29}_{0.002}$ | $0.48_{0.08}$ | $\mathbf{0.29}_{0.019}$ |
| | 70% | $\mathbf{0.30}_{0.002}$ | $\mathbf{0.30}_{0.009}$ | $\underline{0.61}_{0.17}$ | $\mathbf{0.30}_{0.012}$ |
| | 80% | $0.31_{0.004}$ | $\underline{0.29}_{0.006}$ | $0.73_{0.13}$ | $\mathbf{0.28}_{0.010}$ |
| | 90% | $\mathbf{0.32}_{0.005}$ | $\mathbf{0.32}_{0.008}$ | $\underline{1.04}_{0.18}$ | $\mathbf{0.32}_{0.043}$ |
| 720 | 60% | $\underline{0.38}_{0.002}$ | $\underline{0.38}_{0.008}$ | $0.54_{0.07}$ | $\mathbf{0.31}_{0.014}$ |
| | 70% | $\underline{0.38}_{0.002}$ | $\underline{0.38}_{0.008}$ | $0.64_{0.14}$ | $\mathbf{0.31}_{0.013}$ |
| | 80% | $\underline{0.38}_{0.003}$ | $0.39_{0.009}$ | $0.80_{0.16}$ | $\mathbf{0.32}_{0.020}$ |
| | 90% | $\underline{0.39}_{0.006}$ | $0.40_{0.01}$ | $1.00_{0.21}$ | $\mathbf{0.34}_{0.024}$ |

## E.5 Simpler Imputation Methods

We investigate the role of simple imputation strategies, such as Last Observation Carried Forward (LOCF), within the IMTS modeling pipeline for regular MTS methods. As shown in Table 15, on seasonal datasets such as ETTm2, such straightforward methods can at times outperform more complex alternatives. While this might raise questions about the relative benefits of complex imputation methods such as SAITS, a more important consideration is whether trends observed on simpler datasets reliably translate to real-world scenarios. The SAITS paper (Du et al., 2023) addresses this by comparing SAITS and LOCF using a standard baseline (RNN classifier) on the PhysioNet dataset, a challenging real-world benchmark with over 80% missing values, and demonstrates the clear advantage of SAITS (see Table 16). Notably, our proposed model-agnostic approach exhibits highly competitive performance at shorter horizons and shows even greater promise as the forecasting horizon increases.

Table 16: Performance on the PhysioNet dataset (higher is better).

| Method | LOCF | SAITS | MissTSM |
|---|---|---|---|
| Accuracy (%) | $\underline{39.5}$ | 42.7 | **57.84** |

## E.6 Hyper-parameter Tuning of Baseline Imputation Models

To strengthen the the claim on performance gain of MissTSM, we carry out rigorous hyper-parameter tuning of the SAITS imputation model on the ETTh2 dataset under both 70% periodic and 70% MCAR missing scenarios. The hyperparameters and their ranges considered were – `n_layers`: [1,2,3], `d_ffn` = [64, 128, 256] and (`d_model`, `n_heads`) = (64, 1), (128, 2), (256, 4) resulting in 27 total combinations. We report the results of iTransformer and PatchTST in the table below. We first trained SAITS on the 27 combinations. Then we trained the downstream model (iTransformer and PatchTST) on all the imputed files to obtain 27 different error metrics. The best configuration, based on test set results, is selected and reported in the following table. The results of the tuning experiment are presented in Table 17.

The best iTransformer results are obtained with the following SAITS configurations -

**MCAR**: `n_layers=1, d_model=64, d_ffn=128, n_heads=1`

**Periodic**: `n_layers=1, d_model=128, d_ffn=64, n_heads=2`

Table 17: Comparison of MissTSM with iTransformer, and PatchTST trained on HP Tuned SAITS model. Under MCAR and Periodic settings.

| | Horizon | MissTSM | iTransformer (SAITS) | iTransformer (HP tuned SAITS) | PatchTST (SAITS) | PatchTST (HP tuned SAITS) |
|---|---|---|---|---|---|---|
| MCAR | 96 | 0.243 | 0.492 | 0.465 | 0.440 | 0.389 |
| | 192 | 0.259 | 0.545 | 0.513 | 0.484 | 0.434 |
| | 336 | 0.283 | 0.579 | 0.546 | 0.434 | 0.387 |
| | 720 | 0.329 | 0.540 | 0.494 | 0.436 | 0.378 |
| Periodic | 96 | 0.246 | 0.691 | 0.456 | 0.581 | 0.389 |
| | 192 | 0.263 | 0.715 | 0.511 | 0.620 | 0.427 |
| | 336 | 0.301 | 0.763 | 0.552 | 0.592 | 0.394 |
| | 720 | 0.353 | 0.773 | 0.484 | 0.644 | 0.408 |

The best PatchTST results are obtained with the following SAITS configurations –

**MCAR**: `n_layers=3, d_model=256, d_ffn=128, n_heads=4`

**Periodic**: `n_layers=3, d_model=256, d_ffn=64, n_heads=4`

Tuning the SAITS model significantly improved the downstream performance (e.g., MSE dropping from 0.773 to 0.484 for T=720 under periodic masking) for both MCAR and periodic masking strategies. This comparison demonstrates that. However, while the imputation-based baselines do improve with specific tuning, the MissTSM framework still consistently demonstrate competitive performance compared to the newly-optimized baselines across all horizons and on both masking patterns.

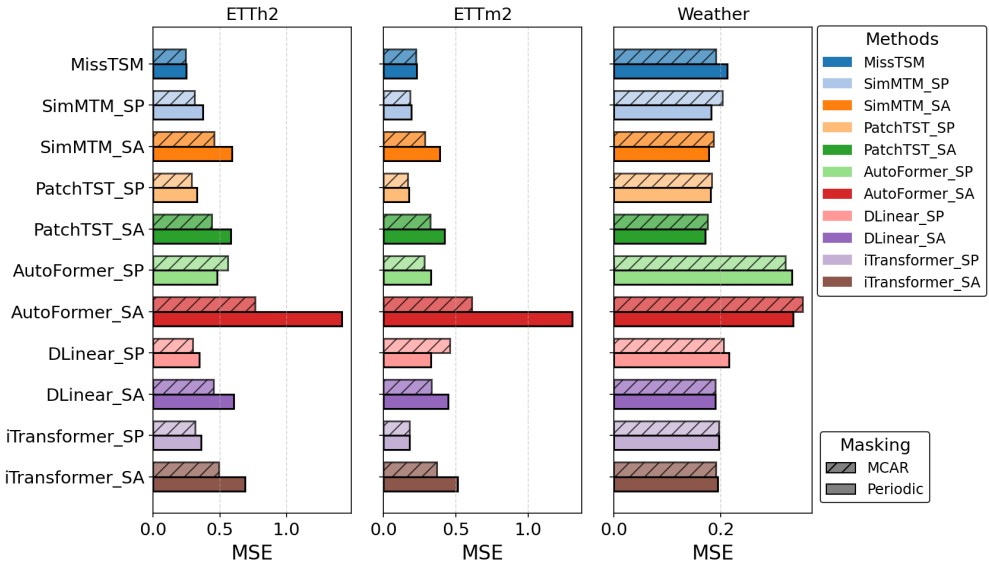

Figure 16: Comparison of different masking methods: MCAR vs. Periodic Masking for ETTh2, ETTm2 and Weather datasets. SA stands for SAITS and SP stands for Spline. Horizon Window: 96. Lower MSE is better

## E.7 Additional Results on Masking strategies

Figures 16, 17 and 18 show additional results on comparison between random and periodic masking strategies for different horizon lengths = 96, 192, 336. Please refer to the main paper for horizon length = 720.

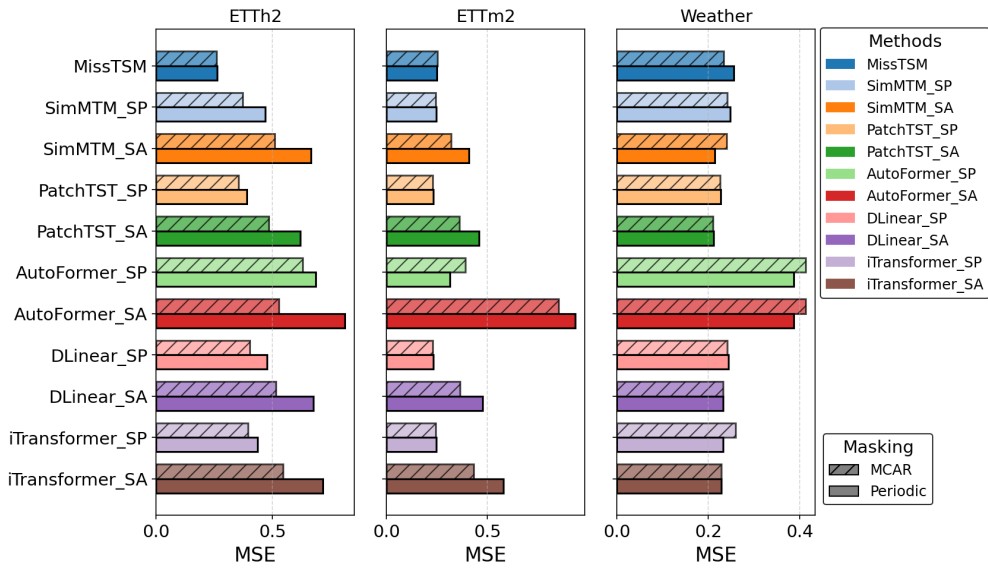

Figure 17: Comparison of different masking methods: MCAR vs. Periodic Masking for ETTh2, ETTm2 and Weather datasets. SA stands for SAITS and SP stands for Spline. Horizon Window: 192. Lower MSE is better

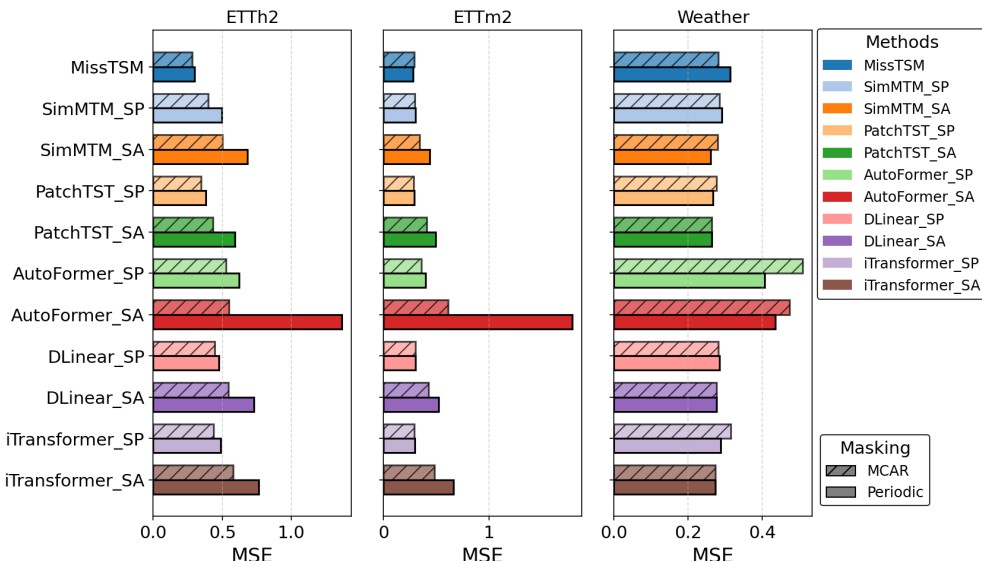

Figure 18: Comparison of different masking methods: MCAR vs. Periodic Masking for ETTh2, ETTm2 and Weather datasets. SA stands for SAITS and SP stands for Spline. Horizon Window: 336. Lower MSE is better

## E.8 Model Agnostic Experiments

### E.8.1 Transformer-based backbones

Tables 18, 19 presents performance of MissTSM integrated with iTransformer, under 60% and 70% masking fractions.

Table 18: MSE (mean$_{std}$) for iTransformer integrated with MissTSM vs with different imputation methods: SAITS and Spline for 60% masking.

| Dataset | Horizon Window | iTransformer + MissTSM | iTransformer + SAITS | iTransformer + Spline |
|---|---|---|---|---|
| ETTh2 | 96 | $\underline{0.350}_{0.006}$ | $0.645_{0.019}$ | $\mathbf{0.325}_{0.013}$ |
| | 192 | $\underline{0.482}_{0.027}$ | $0.612_{0.016}$ | $\mathbf{0.340}_{0.017}$ |
| | 336 | $\mathbf{0.404}_{0.009}$ | $0.526_{0.015}$ | $\underline{0.431}_{0.005}$ |
| | 720 | $\underline{0.490}_{0.012}$ | $0.505_{0.005}$ | $\mathbf{0.436}_{0.017}$ |
| ETTm2 | 96 | $\underline{0.205}_{0.001}$ | $0.361_{0.066}$ | $\mathbf{0.176}_{0.001}$ |
| | 192 | $\underline{0.268}_{0.016}$ | $0.410_{0.064}$ | $\mathbf{0.240}_{0.001}$ |
| | 336 | $\underline{0.325}_{0.018}$ | $0.458_{0.065}$ | $\mathbf{0.290}_{0.002}$ |
| | 720 | $1.080_{0.188}$ | $\underline{0.521}_{0.063}$ | $\mathbf{0.385}_{0.003}$ |
| Weather | 96 | $\mathbf{0.193}_{0.002}$ | $0.194_{0.004}$ | $\underline{0.202}_{0.001}$ |
| | 192 | $\underline{0.240}_{0.004}$ | $\mathbf{0.230}_{0.002}$ | $\underline{0.240}_{0.002}$ |
| | 336 | $0.298_{0.004}$ | $\mathbf{0.280}_{0.001}$ | $\underline{0.291}_{0.001}$ |
| | 720 | $0.364_{0.003}$ | $\mathbf{0.348}_{0.005}$ | $\underline{0.357}_{0.001}$ |

Table 19: MSE (mean$_{std}$) for iTransformer integrated with MissTSM against different imputation methods: SAITS and Spline under 70% masking.

| Dataset | Horizon Window | iTransformer + MissTSM | iTransformer + SAITS | iTransformer + Spline |
|---|---|---|---|---|
| ETTh2 | 96 | $\underline{0.386}_{0.020}$ | $0.702_{0.054}$ | $\mathbf{0.346}_{0.013}$ |
| | 192 | $\underline{0.530}_{0.025}$ | $0.660_{0.058}$ | $\mathbf{0.430}_{0.011}$ |
| | 336 | $\mathbf{0.400}_{0.013}$ | $0.573_{0.064}$ | $\underline{0.463}_{0.026}$ |
| | 720 | $\underline{0.515}_{0.021}$ | $0.575_{0.060}$ | $\mathbf{0.475}_{0.020}$ |
| ETTm2 | 96 | $\underline{0.226}_{0.016}$ | $0.481_{0.130}$ | $\mathbf{0.183}_{0.007}$ |
| | 192 | $\underline{0.300}_{0.027}$ | $0.555_{0.175}$ | $\mathbf{0.244}_{0.008}$ |
| | 336 | $\underline{0.498}_{0.070}$ | $0.592_{0.167}$ | $\mathbf{0.292}_{0.007}$ |
| | 720 | $0.731_{0.590}$ | $\underline{0.650}_{0.158}$ | $\mathbf{0.390}_{0.007}$ |
| Weather | 96 | $\underline{0.188}_{0.012}$ | $\mathbf{0.177}_{0.022}$ | $0.365_{0.255}$ |
| | 192 | $\underline{0.251}_{0.013}$ | $\mathbf{0.218}_{0.019}$ | $0.314_{0.129}$ |
| | 336 | $\underline{0.300}_{0.002}$ | $\mathbf{0.273}_{0.023}$ | $0.346_{0.103}$ |
| | 720 | $\underline{0.368}_{0.003}$ | $\mathbf{0.340}_{0.014}$ | $0.421_{0.112}$ |

### E.8.2 Non-transformer-based backbones

Tables 20 and 21 presents performance of MissTSM integrated with non-transformer backbone - LSTM, under 60% and 70% MCAR and Periodic masking fractions.

Table 20: MSE (mean$_{std}$) for LSTM integrated with MissTSM vs with different imputation methods: SAITS and Spline for 60% masking.

| Dataset | Horizon Window | LSTM + MissTSM | LSTM + SAITS | LSTM + Spline |
|---|---|---|---|---|
| ETTh2 | 96 | $\underline{0.43}_{0.04}$ | $0.53_{0.02}$ | $\mathbf{0.39}_{0.002}$ |
| | 192 | $\underline{0.50}_{0.02}$ | $0.59_{0.03}$ | $\mathbf{0.44}_{0.004}$ |
| | 336 | $\mathbf{0.40}_{0.01}$ | $0.51_{0.09}$ | $\underline{0.42}_{0.02}$ |
| | 720 | $\mathbf{0.44}_{0.01}$ | $0.56_{0.16}$ | $\underline{0.50}_{0.05}$ |
| ETTm2 | 96 | $\underline{0.33}_{0.04}$ | $0.41_{0.09}$ | $\mathbf{0.22}_{0.01}$ |
| | 192 | $0.46_{0.02}$ | $\underline{0.42}_{0.07}$ | $\mathbf{0.27}_{0.01}$ |
| | 336 | $\underline{0.46}_{0.01}$ | $0.49_{0.07}$ | $\mathbf{0.33}_{0.002}$ |
| | 720 | $0.49_{0.06}$ | $\underline{0.47}_{0.02}$ | $\mathbf{0.44}_{0.00}$ |
| Weather | 96 | $\underline{0.34}_{0.005}$ | $\mathbf{0.27}_{0.01}$ | $0.35_{0.02}$ |
| | 192 | $0.38_{0.015}$ | $\mathbf{0.34}_{0.02}$ | $\underline{0.37}_{0.03}$ |
| | 336 | $\mathbf{0.32}_{0.03}$ | $0.42_{0.01}$ | $\underline{0.38}_{0.03}$ |
| | 720 | $\mathbf{0.39}_{0.02}$ | $0.49_{0.06}$ | $\underline{0.45}_{0.00}$ |

Table 21: MSE (mean$_{std}$) for LSTM integrated with MissTSM vs with different imputation methods: SAITS and Spline for 70% masking.

| Dataset | Horizon Window | LSTM + MissTSM | LSTM + SAITS | LSTM + Spline |
|---|---|---|---|---|
| ETTh2 | 96 | $\underline{0.46}_{0.06}$ | $0.51_{0.02}$ | $\mathbf{0.42}_{0.02}$ |
| | 192 | $\underline{0.52}_{0.01}$ | $0.60_{0.06}$ | $\mathbf{0.45}_{0.01}$ |
| | 336 | $\mathbf{0.40}_{0.03}$ | $0.46_{0.02}$ | $\underline{0.42}_{0.01}$ |
| | 720 | $\mathbf{0.46}_{0.01}$ | $\underline{0.48}_{0.00}$ | $0.56_{0.03}$ |
| ETTm2 | 96 | $\underline{0.44}_{0.07}$ | $0.47_{0.01}$ | $\mathbf{0.22}_{0.01}$ |
| | 192 | $\underline{0.51}_{0.02}$ | $0.54_{0.04}$ | $\mathbf{0.30}_{0.02}$ |
| | 336 | $0.60_{0.02}$ | $\underline{0.53}_{0.04}$ | $\mathbf{0.36}_{0.01}$ |
| | 720 | $\underline{0.63}_{0.07}$ | $0.63_{0.15}$ | $\mathbf{0.45}_{0.01}$ |
| Weather | 96 | $0.38_{0.01}$ | $\mathbf{0.26}_{0.07}$ | $\underline{0.28}_{0.01}$ |
| | 192 | $\underline{0.33}_{0.04}$ | $\mathbf{0.32}_{0.04}$ | $0.34_{0.01}$ |
| | 336 | $\underline{0.40}_{0.02}$ | $\mathbf{0.38}_{0.00}$ | $0.44_{0.02}$ |
| | 720 | $\mathbf{0.47}_{0.01}$ | $\underline{0.48}_{0.02}$ | $0.49_{0.04}$ |

Similar to transformer-based backbones, MissTSM, with a non-transformer backbone, consistently achieves competitive performance across datasets and forecasting horizons. We see that Spline imputed model generally performs well on the ETT datasets with fewer variates ($N = 7$) but is not as effective on the Weather dataset, which has $3\times$ as many variates ($N = 21$)

## F Analysis of impact of frequency and phase parameters

In this section, we provide additional details regarding an ablation we conducted to understand the impact of frequency and phase parameters under periodic masking. Given the varying frequency and phase for each feature, we modify the intervals of both to assess their impact on the results. Dataset=ETTh2, Fraction=90%

Table 22: Effect (MSE) of sampling from different frequency intervals. Original refers to original periodic masking MSE. The best results are in bold and second-best are italicized

| Horizon | Original | High Frequency | Low Frequency |
|---|---|---|---|
| 96 | **0.268** $_{0.015}$ | _0.281_ $_{0.028}$ | 0.285 $_{0.023}$ |
| 192 | **0.295** $_{0.029}$ | _0.301_ $_{0.037}$ | 0.316 $_{0.049}$ |
| 336 | 0.319 $_{0.018}$ | _0.308_ $_{0.014}$ | **0.307** $_{0.011}$ |
| 720 | 0.356 $_{0.031}$ | **0.339** $_{0.043}$ | _0.351_ $_{0.058}$ |

Table 23: Effect (MSE) of sampling from different phase intervals. Original refers to the original periodic masking MSE. The best results are in bold and second-best are italicized

| Horizon | Original | (+) Half Cycle | (-) Half Cycle |
|---|---|---|---|
| 96 | **0.268** $_{0.015}$ | _0.287_ $_{0.037}$ | 0.293 $_{0.040}$ |
| 192 | **0.295** $_{0.029}$ | _0.309_ $_{0.050}$ | 0.313 $_{0.057}$ |
| 336 | 0.319 $_{0.018}$ | _0.316_ $_{0.022}$ | **0.311** $_{0.013}$ |
| 720 | 0.356 $_{0.031}$ | _0.343_ $_{0.035}$ | **0.340** $_{0.040}$ |

**Case 1**. With the phase interval held constant, we lower the frequency range and examine two intervals: one in the high frequency region ([0.6, 0.9]) and one in the low frequency region ([0.1, 0.3]). The performance comparison between these new strategies and the original configuration is shown in Table 22.

We observe that with a reduced frequency range, for both high and low frequency intervals, the performance improves as the prediction window increases.

**Case 2**. Following a similar approach as Case 1, we keep the frequency interval constant and lower the range of phase values. We examine the following intervals: the positive half-cycle $[0, \pi]$ and the negative half-cycle $[\pi, 2\pi]$. Table 23 presents the results of this ablation

We observe a similar pattern here as well, with the performance improving as the prediction window increases when we sample from either the positive or negative cycle.

As shown in the tables above, frequency and phase values clearly impact model performance. The new strategies reduce frequency or phase-related randomness among the variates of the dataset, resulting in more consistent values. This appears to enhance the model's ability in long-term forecasting.

## G Case Studies

### G.1 Comparison against noise-agnostic methods

In this section, we compare the proposed MissTSM approach against robust methods like RobustTSF (Cheng et al., 2024) which performs time-series modeling with noisy data, where missing values and anomalies are considered as a type of noise. While Cheng et al. (2024) is an impactful work, there are some key differences compared to MissTSM, such as, (a) RobustTSF treats missing or anomalous values as anomalies and filters them out during training, while MissTSM aim to learn latent representations from all observed time-feature pairs without dropping any time-points, fully leveraging available data; (b) While RobustTSF relies on assumptions like smooth trend filtering and Dirac-weighted anomaly scores with user-set thresholds, which may not hold in practice, MissTSM avoids such assumptions, depending purely on supervision from observed data; (c) RobustTSF assumes no missing values during testing and only imputes zeros if missing. MissTSM robustly handles missing values both during training and testing; (d) While RobustTSF is mainly univariate, MissTSM is explicitly designed for multivariate time-series, capturing dependencies across variates.

Table 24 compares the forecasting performance between MissTSM with MAE vs RobustTSF applied to a MTS model, Autoformer, on the ETTh2 dataset across 60% to 90% MCAR masking fractions.

Table 24: Performance comparison between RobustTSF + Autoformer and MissTSM across fractions and horizons

| Fraction | Horizon | RobustTSF + Autoformer | MissTSM |
|---|---|---|---|
| 60% | 96 | $1.5_{0.054}$ | $\mathbf{0.24}_{0.007}$ |
| | 192 | $1.45_{0.065}$ | $\mathbf{0.26}_{0.002}$ |
| | 336 | $1.34_{0.082}$ | $\mathbf{0.28}_{0.009}$ |
| | 720 | $1.33_{0.024}$ | $\mathbf{0.33}_{0.012}$ |
| 70% | 96 | $1.81_{0.117}$ | $\mathbf{0.25}_{0.009}$ |
| | 192 | $1.76_{0.056}$ | $\mathbf{0.27}_{0.013}$ |
| | 336 | $1.76_{0.049}$ | $\mathbf{0.30}_{0.008}$ |
| | 720 | $1.69_{0.032}$ | $\mathbf{0.35}_{0.010}$ |
| 80% | 96 | $2.2_{0.084}$ | $\mathbf{0.26}_{0.025}$ |
| | 192 | $2.23_{0.091}$ | $\mathbf{0.28}_{0.021}$ |
| | 336 | $2.2_{0.085}$ | $\mathbf{0.29}_{0.029}$ |
| | 720 | $2.11_{0.041}$ | $\mathbf{0.37}_{0.056}$ |
| 90% | 96 | $2.72_{0.077}$ | $\mathbf{0.32}_{0.029}$ |
| | 192 | $2.74_{0.036}$ | $\mathbf{0.35}_{0.072}$ |
| | 336 | $2.77_{0.024}$ | $\mathbf{0.32}_{0.014}$ |
| | 720 | $2.67_{0.034}$ | $\mathbf{0.35}_{0.045}$ |

## G.2 Adapting MissTSM to Anomaly Detection

Table 25: Comparison between RobustTSF + Autoformer and Anomaly Transformer + MissTSM across forecasting horizons, on the PSM dataset

| Horizon | RobustTSF + Autoformer (MSE) | Anomaly Transformer + MissTSM (MSE) |
|---|---|---|
| 96 | **0.349** | 0.360 |
| 192 | 0.524 | **0.465** |
| 336 | 0.731 | **0.603** |
| 720 | 0.958 | **0.815** |

While handling anomalies is not the primary focus of our work, this case study was carried out to explore the possibility of our framework to be adapted to handle other data quality issues such as anomalies. The experiment was carried out on the Pooled Server Metrics (PSM) (Abdulaal et al., 2021) dataset using the following setup - we first use an off-the-shelf anomaly detection method, Anomaly Transformer (Xu et al., 2021), to identify anomalies in the input dataset, which are then treated as missing values in the training of MissTSM. We call this combined framework: Anomaly Transformer + MissTSM. We use 134K samples for training the Anomaly Transformer and use the remaining 84K samples for training and evaluating two forecasting models, namely, Anomaly Transformer + MissTSM, and RobustTSF (with AutoFormer backbone). We specifically split the 84K samples into train, test, and validation sets using a 7:2:1 ratio. Once trained, we evaluate (see Table 25) both MissTSM and RobustTSF on the test set ignoring anomalous points in the forecasting window (using ground-truth labels of anomalies).

# H Visualization

Visualizations on Lake (see Figure 19) and benchmark datasets (see Figures 20, 21, 22, 23, 24)

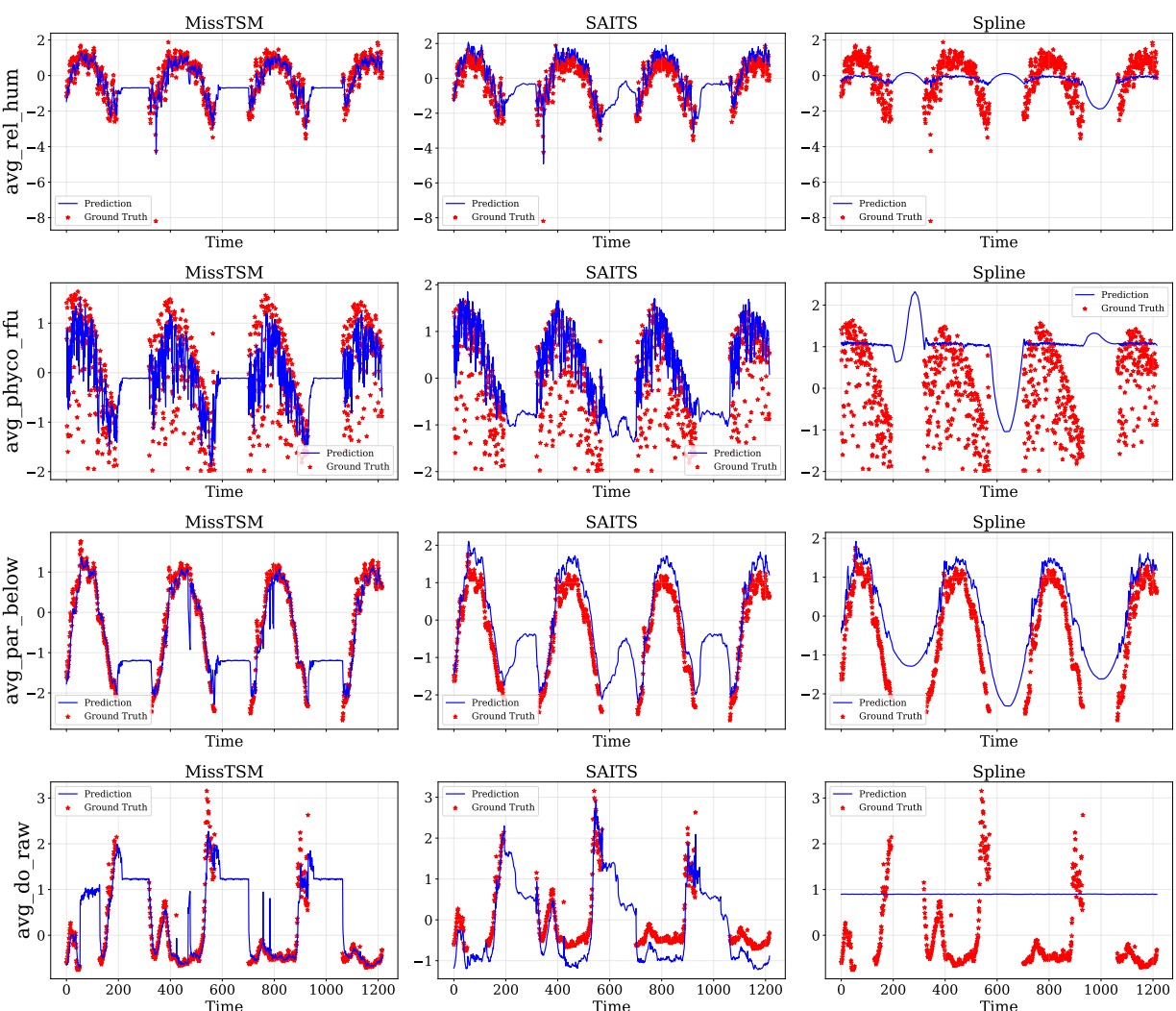

Figure 19: Comparison of T+1 step-ahead predictions on Lake Mendota dataset using three methods for handling missing data: MissTSM, SAITS, and Spline (shown across columns). Blue lines represent model predictions, while red stars indicate the ground truth values. This comparison is based on forecasts generated by the iTransformer model, configured with a prediction length of 28 and a context length of 21. Each row corresponds to a different feature. The selected features include avg_rel_hum, representing average relative humidity with a total of 49.5% missing values; avg_phyco_rfu, which captures average phycocyanin fluorescence and has 48.2% missingness; avg_par_below, measuring photosynthetically active radiation below the water surface with 72.8% of its values missing; and avg_do_raw, representing average raw dissolved oxygen with 48.4% missing data.

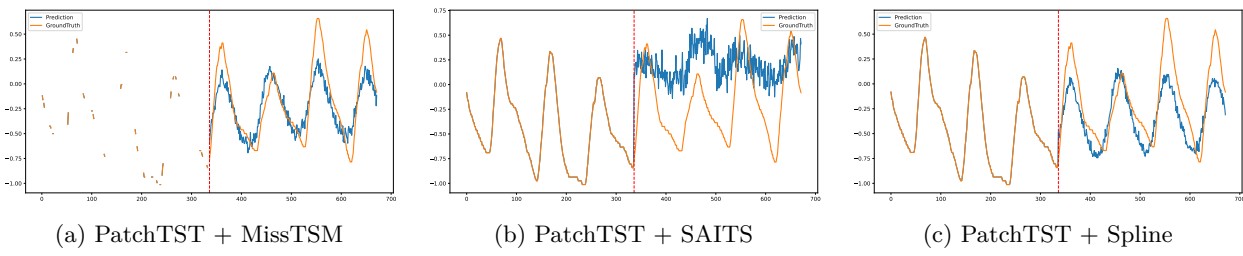

(a) PatchTST + MissTSM     (b) PatchTST + SAITS     (c) PatchTST + Spline

Figure 20: Qualitative comparison on ETTm2 dataset at 70% missing fraction and horizon window = 336.

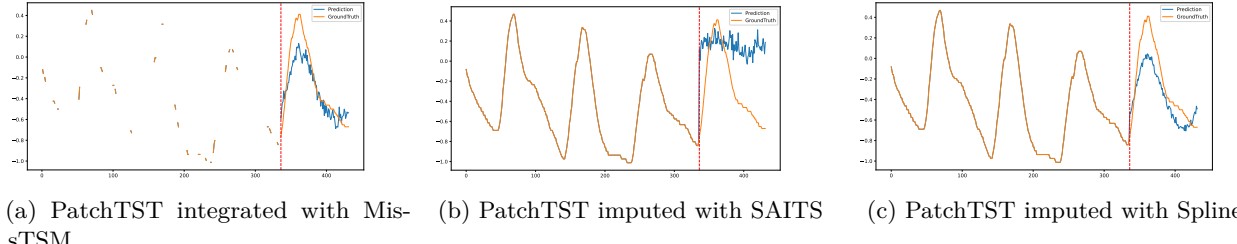

(a) PatchTST integrated with Mis-sTSM    (b) PatchTST imputed with SAITS    (c) PatchTST imputed with Spline

Figure 21: Qualitative comparison on ETTm2 dataset at 70% missing fraction and horizon window=96

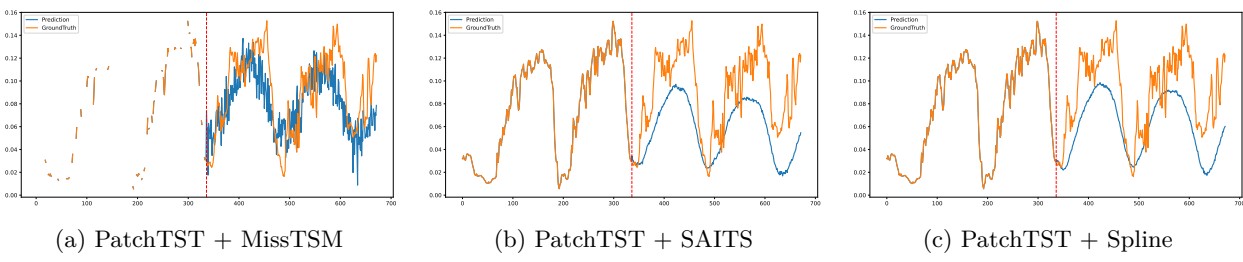

(a) PatchTST + MissTSM     (b) PatchTST + SAITS     (c) PatchTST + Spline

Figure 22: Qualitative comparison on Weather dataset at 60% missing fraction and horizon window = 336.

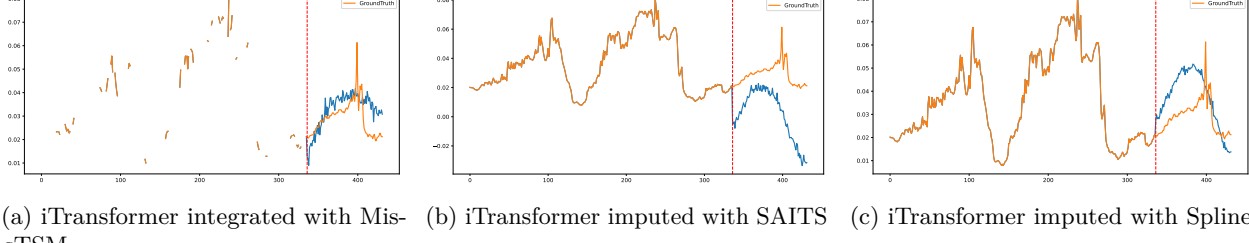

(a) iTransformer integrated with Mis-sTSM    (b) iTransformer imputed with SAITS    (c) iTransformer imputed with Spline

Figure 23: Qualitative comparison on Weather dataset at 60% missing fraction and horizon window=96.

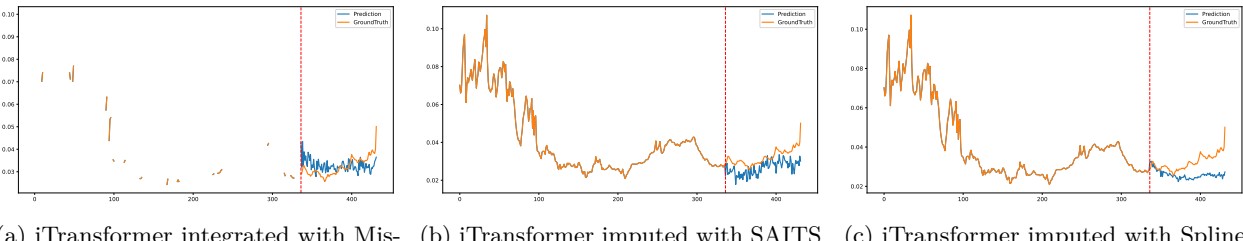

(a) iTransformer integrated with Mis-sTSM    (b) iTransformer imputed with SAITS    (c) iTransformer imputed with Spline

Figure 24: Qualitative comparison on Weather dataset at 70% missing fraction and horizon window=96.

# I   Summary of Results

Table 26 presents the full set of results of all models over different masking fractions, horizons, and masking strategies

Table 26: Comparing forecasting performance of baseline methods using mean squared error (MSE) as the evaluation metric under no masking, MCAR masking, and periodic masking. For every dataset, we consider multiple forecasting horizons, $T \in \{96, 192, 336, 720\}$. Results are color-coded as **Best** , **Second best** . We report the mean and standard deviations (in brackets) across 5 random sampling of the masking schemes (fraction 70%). Subscript $_{SP}$ refer to Spline and $_{SA}$ refer to SAITS

.

| | | ETTh2 | | | | ETTm2 | | | | Weather | | | | Avg |
| | | 96 | 192 | 336 | 720 | 96 | 192 | 336 | 720 | 96 | 192 | 336 | 720 | Rank |
|---|---|---|---|---|---|---|---|---|---|---|---|---|---|---|
| No Masking | MissTSM | 0.255 | 0.234 | 0.316 | 0.305 | 0.183 | 0.209 | 0.261 | 0.311 | 0.164 | 0.210 | 0.254 | 0.324 | 1.9 |
| | SimMTM | 0.295 | 0.356 | 0.375 | 0.404 | 0.172 | 0.223 | 0.282 | 0.374 | 0.163 | 0.203 | 0.255 | 0.326 | 2.9 |
| | PatchTST | 0.274 | 0.338 | 0.330 | 0.378 | 0.164 | 0.220 | 0.277 | 0.367 | 0.151 | 0.196 | 0.249 | 0.319 | 1.7 |
| | AutoFormer | 0.501 | 0.516 | 0.565 | 0.462 | 0.352 | 0.337 | 0.494 | 0.474 | 0.306 | 0.434 | 0.437 | 0.414 | 5.9 |
| | DLinear | 0.288 | 0.383 | 0.447 | 0.605 | 0.168 | 0.224 | 0.299 | 0.414 | 0.175 | 0.219 | 0.265 | 0.323 | 4.1 |
| | iTransformer | 0.304 | 0.392 | 0.425 | 0.415 | 0.176 | 0.246 | 0.289 | 0.379 | 0.163 | 0.203 | 0.256 | 0.326 | 4.5 |
| MCAR Masking | MissTSM | $0.243_{0.006}$ | $0.259_{0.002}$ | $0.283_{0.009}$ | $0.329_{0.011}$ | $0.224_{0.005}$ | $0.253_{0.009}$ | $0.293_{0.019}$ | $0.316_{0.014}$ | $0.191_{0.003}$ | $0.234_{0.006}$ | $0.281_{0.004}$ | $0.322_{0.008}$ | 2.7 |
| | SimMTM$_{SP}$ | $0.309_{0.001}$ | $0.372_{0.005}$ | $0.396_{0.01}$ | $0.418_{0.008}$ | $0.185_{0.001}$ | $0.243_{0.002}$ | $0.298_{0.001}$ | $0.388_{0.005}$ | $0.203_{0.009}$ | $0.242_{0.010}$ | $0.284_{0.008}$ | $0.386_{0.008}$ | 5.0 |
| | SimMTM$_{SA}$ | $0.457_{0.06}$ | $0.510_{0.061}$ | $0.503_{0.055}$ | $0.472_{0.066}$ | $0.287_{0.037}$ | $0.320_{0.035}$ | $0.342_{0.017}$ | $0.413_{0.014}$ | $0.187_{0.002}$ | $0.240_{0.001}$ | $0.280_{0.001}$ | $0.385_{0.004}$ | 6.2 |
| | PatchTST$_{SP}$ | $0.290_{0.003}$ | $0.355_{0.003}$ | $0.345_{0.003}$ | $0.390_{0.003}$ | $0.169_{0.001}$ | $0.228_{0.001}$ | $0.286_{0.001}$ | $0.378_{0.001}$ | $0.183_{0.009}$ | $0.226_{0.009}$ | $0.277_{0.009}$ | $0.339_{0.008}$ | 2.1 |
| | PatchTST$_{SA}$ | $0.440_{0.059}$ | $0.484_{0.057}$ | $0.434_{0.059}$ | $0.436_{0.075}$ | $0.324_{0.05}$ | $0.362_{0.045}$ | $0.410_{0.049}$ | $0.462_{0.047}$ | $0.175_{0.002}$ | $0.211_{0.000}$ | $0.264_{0.002}$ | $0.335_{0.001}$ | 4.6 |
| | AutoFormer$_{SP}$ | $0.559_{0.05}$ | $0.628_{0.101}$ | $0.525_{0.037}$ | $0.550_{0.143}$ | $0.280_{0.006}$ | $0.390_{0.158}$ | $0.360_{0.018}$ | $0.475_{0.033}$ | $0.321_{0.008}$ | $0.413_{0.013}$ | $0.508_{0.036}$ | $0.467_{0.032}$ | 8.9 |
| | AutoFormer$_{SA}$ | $0.767_{0.126}$ | $0.526_{0.06}$ | $0.550_{0.019}$ | $0.449_{0.010}$ | $0.610_{0.312}$ | $0.850_{0.365}$ | $0.615_{0.151}$ | $1.045_{0.262}$ | $0.353_{0.013}$ | $0.413_{0.006}$ | $0.474_{0.028}$ | $0.504_{0.049}$ | 10.2 |
| | DLinear$_{SP}$ | $0.296_{0.003}$ | $0.401_{0.018}$ | $0.445_{0.006}$ | $0.607_{0.013}$ | $0.458_{0.169}$ | $0.228_{0.001}$ | $0.302_{0.000}$ | $0.531_{0.144}$ | $0.205_{0.007}$ | $0.241_{0.007}$ | $0.282_{0.008}$ | $0.373_{0.009}$ | 6.5 |
| | DLinear$_{SA}$ | $0.454_{0.053}$ | $0.514_{0.053}$ | $0.542_{0.064}$ | $0.680_{0.084}$ | $0.330_{0.065}$ | $0.365_{0.062}$ | $0.427_{0.058}$ | $0.538_{0.063}$ | $0.190_{0.001}$ | $0.233_{0.000}$ | $0.276_{0.000}$ | $0.333_{0.001}$ | 6.8 |
| | iTransformer$_{SP}$ | $0.313_{0.004}$ | $0.394_{0.014}$ | $0.436_{0.005}$ | $0.429_{0.005}$ | $0.178_{0.001}$ | $0.243_{0.0004}$ | $0.293_{0.001}$ | $0.384_{0.008}$ | $0.197_{0.006}$ | $0.260_{0.007}$ | $0.315_{0.008}$ | $0.349_{0.006}$ | 4.9 |
| | iTransformer$_{SA}$ | $0.492_{0.058}$ | $0.545_{0.048}$ | $0.579_{0.049}$ | $0.540_{0.094}$ | $0.369_{0.080}$ | $0.432_{0.083}$ | $0.482_{0.083}$ | $0.541_{0.075}$ | $0.191_{0.002}$ | $0.228_{0.002}$ | $0.273_{0.002}$ | $0.348_{0.003}$ | 7.7 |
| Periodic Masking | MissTSM | $0.246_{0.018}$ | $0.263_{0.017}$ | $0.301_{0.042}$ | $0.353_{0.015}$ | $0.227_{0.006}$ | $0.249_{0.006}$ | $0.282_{0.011}$ | $0.337_{0.036}$ | $0.212_{0.007}$ | $0.256_{0.006}$ | $0.313_{0.009}$ | $0.379_{0.019}$ | 4.1 |
| | SimMTM$_{SP}$ | $0.372_{0.122}$ | $0.469_{0.198}$ | $0.496_{0.198}$ | $0.510_{0.200}$ | $0.192_{0.010}$ | $0.247_{0.009}$ | $0.301_{0.008}$ | $0.391_{0.008}$ | $0.182_{0.004}$ | $0.248_{0.003}$ | $0.291_{0.009}$ | $0.344_{0.005}$ | 4.7 |
| | SimMTM$_{SA}$ | $0.591_{0.132}$ | $0.666_{0.152}$ | $0.681_{0.182}$ | $0.667_{0.222}$ | $0.389_{0.071}$ | $0.409_{0.054}$ | $0.436_{0.076}$ | $0.505_{0.055}$ | $0.178_{0.002}$ | $0.214_{0.001}$ | $0.261_{0.001}$ | $0.354_{0.003}$ | 6.0 |
| | PatchTST$_{SP}$ | $0.328_{0.047}$ | $0.389_{0.040}$ | $0.381_{0.050}$ | $0.426_{0.058}$ | $0.174_{0.004}$ | $0.231_{0.003}$ | $0.289_{0.004}$ | $0.381_{0.004}$ | $0.181_{0.004}$ | $0.227_{0.005}$ | $0.267_{0.005}$ | $0.346_{0.003}$ | 2.4 |
| | PatchTST$_{SA}$ | $0.581_{0.120}$ | $0.620_{0.132}$ | $0.592_{0.170}$ | $0.644_{0.230}$ | $0.423_{0.054}$ | $0.457_{0.042}$ | $0.493_{0.037}$ | $0.527_{0.027}$ | $0.171_{0.002}$ | $0.212_{0.001}$ | $0.263_{0.005}$ | $0.334_{0.001}$ | 5.2 |
| | Autoformer$_{SP}$ | $0.482_{0.041}$ | $0.685_{0.165}$ | $0.621_{0.166}$ | $0.546_{0.035}$ | $0.329_{0.109}$ | $0.315_{0.010}$ | $0.398_{0.090}$ | $0.456_{0.021}$ | $0.333_{0.0176}$ | $0.387_{0.035}$ | $0.406_{0.025}$ | $0.453_{0.016}$ | 7.5 |
| | Autoformer$_{SA}$ | $1.415_{0.807}$ | $0.810_{0.269}$ | $1.364_{0.760}$ | $0.820_{0.467}$ | $1.303_{1.278}$ | $0.933_{0.444}$ | $1.788_{0.538}$ | $0.809_{0.431}$ | $0.335_{0.009}$ | $0.387_{0.017}$ | $0.435_{0.035}$ | $0.467_{0.017}$ | 10.8 |
| | DLinear$_{SP}$ | $0.346_{0.069}$ | $0.475_{0.108}$ | $0.477_{0.044}$ | $0.649_{0.068}$ | $0.327_{0.188}$ | $0.230_{0.002}$ | $0.305_{0.003}$ | $0.473_{0.038}$ | $0.215_{0.013}$ | $0.244_{0.013}$ | $0.284_{0.008}$ | $0.339_{0.007}$ | 5.0 |
| | DLinear$_{SA}$ | $0.605_{0.109}$ | $0.674_{0.11}$ | $0.728_{0.138}$ | $0.911_{0.158}$ | $0.447_{0.049}$ | $0.475_{0.043}$ | $0.523_{0.042}$ | $0.626_{0.032}$ | $0.190_{0.001}$ | $0.233_{0.000}$ | $0.276_{0.001}$ | $0.333_{0.001}$ | 7.4 |
| | iTransformer$_{SP}$ | $0.358_{0.070}$ | $0.435_{0.067}$ | $0.488_{0.096}$ | $0.497_{0.119}$ | $0.180_{0.005}$ | $0.245_{0.006}$ | $0.296_{0.007}$ | $0.384_{0.007}$ | $0.197_{0.009}$ | $0.233_{0.006}$ | $0.288_{0.01}$ | $0.351_{0.010}$ | 4.2 |
| | iTransformer$_{SA}$ | $0.691_{0.143}$ | $0.715_{0.140}$ | $0.763_{0.153}$ | $0.773_{0.201}$ | $0.512_{0.055}$ | $0.578_{0.052}$ | $0.662_{0.05}$ | $0.680_{0.029}$ | $0.194_{0.001}$ | $0.229_{0.004}$ | $0.274_{0.002}$ | $0.350_{0.003}$ | 8.2 |

