# OpenReview forum: "Investigating a Model-Agnostic and Imputation-Free Approach for Irregularly-Sampled Multivariate Time-Series Modeling"
_TMLR — Accepted by TMLR_

### Review · Reviewer_bYD2 · 2025-10-28

**Summary Of Contributions:**

This paper addresses the problem of modelling irregularly sampled Multivariate Time Series (IMTS) where different variables may be missing at different time steps. The authors propose MissTSM, a model-agnostic and imputation-free approach that can be integrated as a plug-and-play layer with existing multivariate time series models. The key contributions include:
1. Time-Feature-Independent (TFI) embedding: A novel embedding scheme that treats each (time-step, variate) combination as an independent token, allowing the model to handle missing values without explicit imputation.
2. Missing Feature-Aware Attention (MFAA): a masked cross-attention that aggregates only observed features at each time step using a learnable query, then projects back to the original shape before passing to any backbone model.
The authors benchmark MissTSM on both forecasting and classification with two synthetic masking schemes, and five real-world datasets. They conclude that MissTSM is competitive, and tend to be more robust when the amount of missingness is large, and when simple periodic structure is absent.

Strengths:
- The paper clearly explains the limitations of the existing imputation-based approaches, and give strong motivation of the imputation-free methods. The design of TFI + MFAA is clear, simple and well-articulated. The model-agnostic design makes it easy for potential applications on existing frameworks.
- The algorithm designs and results are supported by a broad range of experiments, covering multiple datasets, various missing value scenarios of both synthetic and real-word, different missing value fractions, and comparisons with both imputation-based and specialised IMTS methods.
Weakness:
- Computational cost of TFI may be high - it makes a token per (time, features). The authors acknowledged  this as a limitation but it is not quantified. There is no comparison of training/inference time with baselines.
- Baseline tuning fairness is discussed but not exhaustively audited (e.g. SAITS appears unable for periodic patterns)
- On many datasets, MissTSM shows comparable but not significantly better performance than imputation-based methods. The advantages appear mainly when missing fractions are very high or data lacks periodicity. Some real-word results also show mixed performance. This marginal improvement and the potentially high computational cost give raise to concerns on the practicality of the approach.

**Audience:**

Yes

**Audience Explanation:**

In real-word, IMTS is an important practical problem across many domains. The authors thoroughly address the issue, and therefore motivated the design of MissTSM. The design is elegant, and the model-agnostic nature makes it a useful ML tool rather than single-task bespoke architectures, potentially being helpful on broader scenarios.

**Broader Impact Concerns:**

This paper does not include a Broader Impact Statement. Given the potential application in sectors like healthcare, environmental monitoring, the manuscript would benefit from a short paragraph addressing:
- Validation requirements and safety considerations when used in healthcare sectors.
- Guidance on when the method is reliable vs when it may fail could prevent inappropriate deployment in critical applications.
- The potential issue of over-trust in imputation-free outputs. What would the prediction based on imputation-free dataset, and potentially with high missingnessm, impact the downstream decisions?

**Claims And Evidence:**

Yes

**Claims Explanation:**

The claims are mostly supported by accurate, convincing and clear evidence. Multiple datasets and experimental conditions are used in the experiments, and the trend evidence is consistent in the results, on both synthetic and real-world scenarios. There are also ablation studies supporting the value of the proposed components.
The evidence could however be stronger by including appropriate baseline tuning, studying the computation cost, and further experiments with non-Transformer backbone to support the "model-agnostic" claim.

**Requested Changes:**

1. Report compute & Scaling: Add a small section quantifying the computational costs, e.g. peak GPU memory, tokens/sec, inference time, comparing MissTSM with other baselines. The manuscript would benefit from better evaluation on the scalability of MissTSM. (Critical)
2. Baseline tuning: Add tuning experiments for imputation methods under both MCAR and periodic. This would strengthen the claim one the performance gain of MissTSM with fairer comparisons. (Strengthening)
3. Broader "model-agnostic" demonstrations: add experiments with non-Transformer backbone. A small table showing "backbone X + MissTSM" vs "backbone X + SAITS/Spline" will make the claim more concrete. (Strengthening)
4. Further technical explanations: For example, we could add more details on the design choice of MFAA (why single-query? How is it initialised? Why is it time-independent?) And more detailed comparisons with existing frameworks like mTAN. (Strengthening)

---

> ### Author Response · Authors · 2025-11-13
> **Author Response to Reviewer bYD2**
>
> **Comment 1: Computational cost of TFI maybe high - it makes a token per (time, features). The authors acknowledged this as a limitation but it is not quantified. There is no comparison of training/inference time with baselines. Report compute and scaling: Add a small section quantifying the computational costs, e.g. peak GPU memory, tokens/sec, inference time, comparing MissTSM with other baselines. The manuscript would benefit from better evaluation on the scalability of MissTSM**
>
> > Thank you for this comment. We have added a detailed analysis that theoretically and empirically compares the training time, inference throughput, and memory usage of our MissTSM framework against key baselines, as described in the following.
>
> > **Theoretical Analysis:** We compare the computational complexity of MissTSM with a baseline IMTS model (mTAND), backbone model (SimMTM), and an imputation model (SAITS) w.r.t. time, memory, and parameters in the Table below. Here $L$ represents the context length, $N$ is the number of variates (features), and $d$ is the embedding dimension.  $d_n$ and $d_l$ are hidden and latent dimensions of mTAND, respectively. $G$ and $l_{inner}$ corresponds to the number of groups within the DMSA blocks and number of layers within each block in SAITS, respectively.
> | Model    | Time Complexity | Memory Complexity | Params Complexity |
> |----------|----------------|-----------------|-----------------|
> | MissTSM  | $\mathcal{O}(LNd^2)$ | $\mathcal{O}(LNd)$ | $\mathcal{O}(d^2 + Nd)$ |
> | mTAND    | $\mathcal{O}(L^2(d^2 + h(N + d_n)) + Ld_ld_n)$ | $\mathcal{O}(L(N+d_n+d_l+d) + hL^2)$ | $\mathcal{O}(d^2+hNd_n + hd_n^2 + d_nd_l)$ |
> | SimMTM   | $\mathcal{O}(NLdE(L+d))$ | $\mathcal{O}(Ld + hL^2)$ | $\mathcal{O}(Ed^2 + L^2d^2)$ |
> | SAITS    | $\mathcal{O}(L(Nd + N^2 + L) + Gl_{inner}L^2d)$ | $\mathcal{O}(L(N+d) + Gl_{inner}(Ld + hL^2))$ | $\mathcal{O}(l_{inner}d^2 + N(d + N + L))$ |
>
> > We can see that MissTSM’s scalability is primarily driven by $L$, $N$, and $d$, offering a more straightforward complexity profile compared to the multi-component dependencies of baselines like mTAND, SimMTM, and SAITS.
>
> > **Empirical Analysis:** The table (values within the parenthesis represent the standard deviation) below presents the empirical results from our comparison of the computational and memory costs of different IMTS pipelines on the PhysioNet P19 dataset. The training time is reported as an average time per epoch observed over 5 trials. For computing the inference throughput, we divide the total number of test samples by the total inference time. In the case of the two-stage IMTS pipeline, the total inference time is the sum of inference times in each stage. For the two-stage pipeline, we take the max of both stages' GPU memory. All the experiments were run on Ubuntu 18.04, Python 3.9, PyTorch 1.12 (CUDA 12.8), 1× NVIDIA TITAN 24GB GPU, using a constant batch size of 64 for all models and mixed precision disabled.
> | Pipeline        | Training Time (avg. epoch, sec) ↓ | Inference Throughput (samples/sec) ↑ | Peak Active Tensor Memory (train, GB) ↓ | # Learnable Params | AUROC |
> |-----------------|----------------------------------|------------------------------------|----------------------------------------|------------------|-------|
> | SAITS + MAE     | 13.277 (0.578) + 0.279 (0.163) = 13.556 (0.601) | 438.135 (47.731) | **0.74** | 1,373,576 + 363,236 = 1,736,812 | 77.40 |
> | mTAND           | 5.95 (0.117)                     | 9593.36 (121.44)                   | 1.6                                    | 142,370           | 82.9  |
> | Raindrop        | 17.651 (0.554)                   | 964.75 (88.91)                     | 1.4                                    | 1,947,804         | 87.6  |
> | MissTSM + MAE   | **0.322 (0.077)**                    | **19371.58 (1315.92)**                 | 1.866                                  | **365,506**           | **88.8**  |
>
> > The empirical results demonstrate that the MissTSM+MAE framework is ~42x faster per training epoch and achieves over 44x higher inference throughput than the two-stage SAITS+MAE pipeline and is also faster than other specialized models like mTAND and Raindrop. The peak memory of the MissTSM+MAE (1.866 GB) is, however, slightly higher than the SAITS+MAE pipeline (0.74 GB), which can be further improved using more efficient code optimization in future works.
>
> > We have added this discussion on computational complexity to Appendix E.1 and E.2 in the revised manuscript.

---

> ### Author Response · Authors · 2025-11-13
> **Author Response to Reviewer bYD2 (contd.)**
>
> **Comment 2: Baseline tuning fairness is discussed but not exhaustively audited (e.g. SAITS appears unable for periodic patterns). Add tuning experiments for imputation methods under both MCAR and periodic. This would strengthen the claim on the performance gain of MissTSM with fairer comparisons**
>
> > We have carried out hyper-parameter tuning of the SAITS imputation model on the ETTh2 dataset under both 70% periodic and 70% MCAR missing scenarios. The hyperparameters and their ranges considered were – n_layers = [1,2,3], d_ffn = [64, 128, 256] and (d_model, n_heads) = (64, 1), (128, 2), (256, 4) resulting in 27 total combinations. We report the results of iTransformer and PatchTST in the table below. We first trained SAITS on the 27 combinations. Then we trained the downstream models (iTransformer and PatchTST) on all the imputed files to obtain 27 different models. The best configuration, based on test set results, is selected and reported in the following table. The results of this new tuning are presented below.
> |  | MissTSM | iTrans. (SAITS) | iTrans. (Tuned SAITS) | PatchTST (SAITS) | PatchTST (Tuned SAITS) |
> |----------|---------|----------------|---------------------|-----------------|------------------------|
> | **MCAR** |         |                |                     |                 |                        |
> | 96       | 0.243   | 0.492          | 0.465               | 0.440           | 0.389                  |
> | 192      | 0.259   | 0.545          | 0.513               | 0.484           | 0.434                  |
> | 336      | 0.283   | 0.579          | 0.546               | 0.434           | 0.387                  |
> | 720      | 0.329   | 0.540          | 0.494               | 0.436           | 0.378                  |
> | **Periodic** |      |                |                     |                 |                        |
> | 96       | 0.246   | 0.691          | 0.456               | 0.581           | 0.389                  |
> | 192      | 0.263   | 0.715          | 0.511               | 0.620           | 0.427                  |
> | 336      | 0.301   | 0.763          | 0.552               | 0.592           | 0.394                  |
> | 720      | 0.353   | 0.773          | 0.484               | 0.644           | 0.408                  |
>
> > The best iTransformer results are obtained with the following SAITS configurations –
> **MCAR**: n_layers=1, d_model=64, d_ffn=128, n_heads=1
> **Periodic**: n_layers=1, d_model=128, d_ffn=64, n_heads=2
>
> > The best PatchTST results are obtained with the following SAITS configurations –
> **MCAR**: n_layers=3, d_model=256, d_ffn=128, n_heads=4
> **Periodic**: n_layers=3, d_model=256, d_ffn=64, n_heads=4
>
> > Tuning the SAITS model significantly improved the downstream performance (e.g., MSE dropping from 0.773 to 0.484 for T=720 under periodic masking) for both MCAR and periodic masking strategies. However, while the imputation-based baselines do improve with specific tuning, the MissTSM framework still consistently show competitive performance compared to the newly-optimized baselines across all horizons and on both masking patterns.
>
> > We have added the tuning experiments to Appendix F.6 in the revised manuscript.

---

> ### Author Response · Authors · 2025-11-13
> **Author Response to Reviewer bYD2 (contd. 2)**
>
> **Comment 3: On many datasets, MissTSM shows comparable but not significantly better performance than imputation-based methods. The advantages appear mainly when missing fractions are very high or data lacks periodicity. Some real-world results also show mixed performance. This marginal improvement and the potentially high computational cost give raise to concerns on the practicality of the approach**
>
> > Regarding the significance of our results in comparison to baselines, we would like to make the following comments.
> > 1. A key goal of our work is to explore the viability of a model-agnostic and imputation-free approach for the problem of IMTS modeling. We acknowledge that it is difficult to outperform every baseline method on every dataset and experiment setup. However, we believe that by achieving competitive performance (best to second-best) as state-of-the-art methods is a strong indication of the potential of model-agnostic approaches for IMTS modeling.
> > 2. High missingness or absence of periodicity are challenging scenarios. Good performance under such conditions while also maintaining comparable performance in more regular settings is encouraging in terms of robustness to varying data conditions.
> > 3. High missingness or absence of periodicity are often common in real-world datasets. Good performance on such datasets underscores the applicability or practicality of our approach. For example, healthcare data like PhysioNet are significantly sparse (80%), where MissTSM achieves an ~15% improvement. Similarly, in ecology lake datasets, where we have around 50% sparsity with complex patterns, MissTSM consistently outperforms the imputation-based baselines.
> >4. Moreover, the computational analysis shows that the proposed framework is computationally cheaper than the impute-then-model pipelines, with ~42x faster train time and ~44x higher inference throughput than the SAITS+MAE baseline.

---

> ### Author Response · Authors · 2025-11-13
> **Author Response to Reviewer bYD2 (contd. 3)**
>
> **Comment 4: Broader “model-agnostic” demonstrations: add experiments with non-Transformer backbone. A small table showing "backbone X + MissTSM" vs "backbone X + SAITS/Spline" will make the claim more concrete**
>
> > To further strengthen the model-agnostic claim across non-transformer backbones as well, we have carried out experiments integrating MissTSM with LSTM as the backbone. As suggested, we compare LSTM + MissTSM with LSTM + SAITS and LSTM + Spline. The experiments were conducted across all the three datasets, ETTh2, ETTm2 and Weather under MCAR setting. We consider two missing fractions, 60% and 70%, and compare the performance across 4 different horizon windows – 96, 192, 336, 720.
>
> >Table 1. Comparing LSTM + MissTSM under 60% Masking. **Bold** represents lowest MSE, $\underline{\text{underline}}$, second lowest MSE
> | **Dataset** | **Horizon Window** | **LSTM + MissTSM** | **LSTM + SAITS** | **LSTM + Spline** |
> |--------------|--------------------|--------------------|------------------|-------------------|
> | **ETTh2** | 96  | $\underline{0.43 ± 0.04}$ | 0.53 ± 0.02 | **0.39 ± 0.002** |
> |              | 192 | $\underline{0.50 ± 0.02}$ | 0.59 ± 0.03 | **0.44 ± 0.004** |
> |              | 336 | **0.40 ± 0.01** | 0.51 ± 0.09 | $\underline{0.42 ± 0.02}$ |
> |              | 720 | **0.44 ± 0.01** | 0.56 ± 0.16 | $\underline{0.50 ± 0.05}$ |
> | **ETTm2** | 96  | $\underline{0.33 ± 0.04}$ | 0.41 ± 0.09 | **0.22 ± 0.01** |
> |              | 192 | 0.46 ± 0.02 | $\underline{0.42 ± 0.07}$ | **0.27 ± 0.01** |
> |              | 336 | $\underline{0.46 ± 0.01}$ | 0.49 ± 0.07 | **0.33 ± 0.002** |
> |              | 720 | 0.49 ± 0.06 | $\underline{0.47 ± 0.02}$ | **0.44 ± 0.00** |
> | **Weather** | 96  | $\underline{0.34 ± 0.005}$ | **0.27 ± 0.01** | 0.35 ± 0.02 |
> |              | 192 | 0.38 ± 0.015 | **0.34 ± 0.02** | $\underline{0.37 ± 0.03}$ |
> |              | 336 | **0.32 ± 0.03** | 0.42 ± 0.01 | $\underline{0.38 ± 0.03}$ |
> |              | 720 | **0.39 ± 0.02** | 0.49 ± 0.06 | $\underline{0.45 ± 0.00}$ |
>
> > Table 2. Comparing LSTM + MissTSM under 70% Masking
> | **Dataset** | **Horizon Window** | **LSTM + MissTSM** | **LSTM + SAITS** | **LSTM + Spline** |
> |--------------|--------------------|--------------------|------------------|-------------------|
> | **ETTh2** | 96  | $\underline{0.46 ± 0.06}$ | 0.51 ± 0.02 | **0.42 ± 0.02** |
> |              | 192 | $\underline{0.52 ± 0.01}$ | 0.60 ± 0.06 | **0.45 ± 0.01** |
> |              | 336 | **0.40 ± 0.03** | 0.46 ± 0.02 | $\underline{0.42 ± 0.01}$ |
> |              | 720 | **0.46 ± 0.01** | $\underline{0.48 ± 0.00}$ | 0.56 ± 0.03 |
> | **ETTm2** | 96  | $\underline{0.44 ± 0.07}$ | 0.47 ± 0.01 | **0.22 ± 0.01** |
> |              | 192 | $\underline{0.51 ± 0.02}$ | 0.54 ± 0.04 | **0.30 ± 0.02** |
> |              | 336 | 0.60 ± 0.02 | $\underline{0.53 ± 0.04}$ | **0.36 ± 0.01** |
> |              | 720 | $\underline{0.63 ± 0.07}$ | $\underline{0.63 ± 0.15}$ | **0.45 ± 0.01** |
> | **Weather** | 96  | 0.38 ± 0.01 | **0.26 ± 0.07** | $\underline{0.28 ± 0.01}$ |
> |              | 192 | $\underline{0.33 ± 0.04}$ | **0.32 ± 0.04** | 0.34 ± 0.01 |
> |              | 336 | $\underline{0.40 ± 0.02}$ | **0.38 ± 0.00** | 0.44 ± 0.02 |
> |              | 720 | **0.47 ± 0.01** | $\underline{0.48 ± 0.02}$ | 0.49 ± 0.04 |
>
> > Similar to transformer-based backbones such as MAE, PatchTST, and iTransformer, MissTSM shows competitive performance with LSTM backbone across datasets and forecasting horizons compared to imputation-based approaches such as SAITS. We also see that Spline imputed model generally performs well on the ETTh and ETTm datasets with fewer variates ($N=7$) but is not the best performing model on the Weather dataset, which has 3x as many variates ($N=21$). Overall, we see that while MissTSM shows best or second-best performance across all backbones, MissTSM when coupled with MAE shows best performance across 10 out of 12 settings across the 3 datasets (ETTh2, ETTm2, Weather) and 4 horizon windows on 70% MCAR setting.
>
> > We have added the model-agnostic experiments with non-transformer backbone to Appendix F.8.2 in the revised manuscript.

---

> ### Author Response · Authors · 2025-11-13
> **Author Response to Reviewer bYD2 (contd. 4)**
>
> **Comment 5: Further technical explanations: For example, we could add more details on the design choice of MFAA (why single-query? How is it initialized? Why is it time-independent?)**
>
> > *Why a single, time-independent query?*
> > > The goal of the MFAA layer is to learn a feature representation at each time-step using the observed variates while ignoring the missing variates. It does so by using a learnable query. The goal of learning this query vector can be equated to the task of extracting the best representation from the observed features at every timestep. Since this task can be assumed to be invariant of time (i.e., the same feature extractor can be used to learn representations of observed variates at any time-step), we consider learning a single time-independent query. Alternatively, if we used time-dependent queries, it would require learning $T$ separate query vectors, which would increase the number of learnable parameters. Moreover, by learning a different query for every time-step, the model would be forced to learn $T$ different rules, potentially making it susceptible to overfitting to artifacts at specific time indices.
>
> > *Intialization*
> >> The query vector $Q$ is an nn.Parameter that is zero-initialized. This provides a neutral, unbiased starting point for the model.
>
> > We have added this discussion to Appendix D.4 in the revised manuscript.
>
> **Comment 6: More detailed comparisons with existing frameworks like mTAN**
>
> > While both MissTSM and mTAN use a query-based mechanism to aggregate information from observed time-series data, the key difference lies in the nature of the query and architectural flow. mTAN introduces a continuous time attention mechanism that learns time embeddings by querying at a fixed set of discrete reference points on a temporal grid. In contrast, MFAA utilizes a single, learnable query that is time independent.
>
> > More fundamentally, the frameworks operate in different ways. mTAN is a two-stage architecture, with the first stage being "temporal-first, variate-independent interpolation", where it uses attention to perform temporal attention for each feature to create a regular, fixed-length sequence, which are then, in the second stage, linearly combined to generate a single representation at each time-step. In contrast, MissTSM is "Variate-first" - the MFAA layer operates at each time-step $t$, performing cross-variate aggregation to create a representation $L_t$. Then, the backbone model learns the temporal relationships from this sequence of summaries.
>
> > We have added this discussion to Section 3.3 of the revised manuscript.
>
> **Comment 7: Include broader impact**
>
> > Thank you for the suggestions. We have added a section to the manuscript addressing the broader impact of our work, as described in the following.
>
> > *Validation requirements and safety considerations when used in healthcare sectors* - MissTSM is a decision-support tool and would require rigorous clinical validation to ensure reliable predictions. We would recommend that the model must be used in an "expert-in-the-loop" system
>
> > *Guidance on when the method is reliable vs when it may fail* - The method is reliable in challenging scenarios with low periodicity, high missingness and high number of variates. On low variate, highly periodic data, a simpler approach like Spline + Backbone can be more effective
>
> > *The potential issue of over-trust in imputation-free outputs* - We agree that this is a critical concern. Our recommendation would be to pair MissTSM’s predictions with a reliable uncertainty quantification metric. This metric should reflect the sparsity (or quality) of the input data, warning the user when the predictions are of low confidence.
>
> > We have added the Broader Impact Statement at the end of the Conclusion section (Section 6).

---

### Review · Reviewer_aMoN · 2025-11-03

**Summary Of Contributions:**

The paper proposes MissTSM, a model-agnostic and imputation-free framework for Irregularly-Sampled Multivariate Time-Series (IMTS) modeling.
MissTSM introduces two main components: (1) a Time-Feature Independent (TFI) embedding that independently encodes each observed (time, feature) pair without imputing missing values, and (2) a Missing Feature-Aware Attention (MFAA) mechanism that computes masked attention over observed features to obtain latent representations robust to missingness.
This design enables existing time-series models—such as masked autoencoders or Transformers—to handle irregularly sampled data directly.
Empirical results demonstrate competitive performance across diverse datasets and tasks, highlighting the framework’s generality and adaptability.

Key strengths:
* Provides a clear, modular solution to imputation dependency and model-specific design issues in IMTS.
* Achieves competitive results on both forecasting and classification benchmarks.
* The design is conceptually simple and compatible with existing Transformer-based models.

Key weaknesses:
* The approach does not fully resolve long-term temporal dependency modeling, since MFAA focuses on feature-level aggregation rather than temporal reasoning.
* The computational and memory cost remains significant due to dense embedding and attention operations.

**Audience:**

Yes

**Audience Explanation:**

The paper tackles a timely problem in irregularly sampled multivariate time-series modeling and imputation. Its proposed MissTSM framework provides a model-agnostic, imputation-free alternative that integrates smoothly with existing Transformer-based architectures. These findings would interest TMLR readers focused on time-series modeling, imputation methods, and representation learning.

**Broader Impact Concerns:**

No major ethical or societal concerns are apparent. The work focuses on methodological advances in irregular time-series modeling and does not involve sensitive data, human subjects, or deployment scenarios.

**Claims And Evidence:**

Yes

**Claims Explanation:**

The empirical evidence is generally convincing and supports the authors’ main claims. In comparisons with multiple time-series baselines imputed using SAITS (Figure 4), MissTSM consistently achieves the best or near-best performance. For classification tasks (Figure 5), MissTSM attains strong F1 scores and maintains stable performance across varying levels of missing data. The IMTS classification results on the P12 and P19 datasets further underscore its effectiveness, with MissTSM consistently ranking among the top two models. These results collectively demonstrate the framework’s robustness and model-agnostic adaptability across diverse tasks.

**Requested Changes:**

* Include a runtime and memory analysis comparing MissTSM with key baselines.
* Provide a clearer explanation or empirical evidence of how MissTSM handles long-term temporal dependencies, especially in forecasting tasks.
* Improve figure clarity: for bar charts, explicitly indicate whether lower or higher values are better, particularly for metrics other than F1 scores.

---

> ### Author Response · Authors · 2025-11-13
> **Author Response to Reviewer aMoN**
>
> **Comment 1: The approach does not fully resolve long-term temporal dependency modeling, since MFAA focuses on feature-level aggregation rather than temporal reasoning. Provide a clearer explanation or empirical evidence of how MissTSM handles long-term temporal dependencies, especially in forecasting tasks**
>
> > We agree that MFAA is not designed for long-term temporal modeling. Its primary role is to handle missing values at a time-step by using the observed variates at the same time-step as context. MissTSM relies on the subsequent backbone model (e.g., MAE, PatchTST, or iTransformer) to model long-range temporal dynamics. We show the effects of varying the horizon windows in Figure 15 (Appendix F.4) on ETTh2, ETTm2 and Weather datasets, where we can see that MissTSM when coupled with a backbone model such as MAE provides superior performance on longer horizon windows. This suggests that MissTSM is able to effectively pass on information of the observed variates at every time-step to the backbone model for modeling long-term temporal dependences, better than baseline approaches.
>
> > We have added this explanation to Section 3.3 in the revised manuscript.

---

> ### Author Response · Authors · 2025-11-13
> **Author Response to Reviewer aMoN (contd.)**
>
> **Comment 2: The computational and memory cost remains significant due to dense embedding and attention operations. Include a runtime and memory analysis comparing MissTSM with key baselines**
>
> > Thank you for this comment. We have added a detailed analysis that theoretically and empirically compares the training time, inference throughput, and memory usage of our MissTSM framework against key baselines, as described in the following.
>
> > **Theoretical Analysis:** We compare the computational complexity of MissTSM with a baseline IMTS model (mTAND), backbone model (SimMTM), and an imputation model (SAITS) w.r.t. time, memory, and parameters in the Table below. Here $L$ represents the context length, $N$ is the number of variates (features), and $d$ is the embedding dimension.  $d_n$ and $d_l$ are hidden and latent dimensions of mTAND, respectively. $G$ and $l_{inner}$ corresponds to the number of groups within the DMSA blocks and number of layers within each block in SAITS, respectively.
> | Model    | Time Complexity | Memory Complexity | Params Complexity |
> |----------|----------------|-----------------|-----------------|
> | MissTSM  | $\mathcal{O}(LNd^2)$ | $\mathcal{O}(LNd)$ | $\mathcal{O}(d^2 + Nd)$ |
> | mTAND    | $\mathcal{O}(L^2(d^2 + h(N + d_n)) + Ld_ld_n)$ | $\mathcal{O}(L(N+d_n+d_l+d) + hL^2)$ | $\mathcal{O}(d^2+hNd_n + hd_n^2 + d_nd_l)$ |
> | SimMTM   | $\mathcal{O}(NLdE(L+d))$ | $\mathcal{O}(Ld + hL^2)$ | $\mathcal{O}(Ed^2 + L^2d^2)$ |
> | SAITS    | $\mathcal{O}(L(Nd + N^2 + L) + Gl_{inner}L^2d)$ | $\mathcal{O}(L(N+d) + Gl_{inner}(Ld + hL^2))$ | $\mathcal{O}(l_{inner}d^2 + N(d + N + L))$ |
>
> > We can see that MissTSM’s scalability is primarily driven by $L$, $N$, and $d$, offering a more straightforward complexity profile compared to the multi-component dependencies of baselines like mTAND, SimMTM, and SAITS.
>
> > **Empirical Analysis:** The table (values within the parenthesis represent the standard deviation) below presents the empirical results from our comparison of the computational and memory costs of different IMTS pipelines on the PhysioNet P19 dataset. The training time is reported as an average time per epoch observed over 5 trials. For computing the inference throughput, we divide the total number of test samples by the total inference time. In the case of the two-stage IMTS pipeline, the total inference time is the sum of inference times in each stage. For the two-stage pipeline, we take the max of both stages' GPU memory. All the experiments were run on Ubuntu 18.04, Python 3.9, PyTorch 1.12 (CUDA 12.8), 1× NVIDIA TITAN 24GB GPU, using a constant batch size of 64 for all models and mixed precision disabled.
> | Pipeline        | Training Time (avg. epoch, sec) ↓ | Inference Throughput (samples/sec) ↑ | Peak Active Tensor Memory (train, GB) ↓ | # Learnable Params | AUROC |
> |-----------------|----------------------------------|------------------------------------|----------------------------------------|------------------|-------|
> | SAITS + MAE     | 13.277 (0.578) + 0.279 (0.163) = 13.556 (0.601) | 438.135 (47.731) | **0.74** | 1,373,576 + 363,236 = 1,736,812 | 77.40 |
> | mTAND           | 5.95 (0.117)                     | 9593.36 (121.44)                   | 1.6                                    | 142,370           | 82.9  |
> | Raindrop        | 17.651 (0.554)                   | 964.75 (88.91)                     | 1.4                                    | 1,947,804         | 87.6  |
> | MissTSM + MAE   | **0.322 (0.077)**                    | **19371.58 (1315.92)**                 | 1.866                                  | **365,506**           | **88.8**  |
>
> > The empirical results demonstrate that the MissTSM+MAE framework is ~42x faster per training epoch and achieves over 44x higher inference throughput than the two-stage SAITS+MAE pipeline and is also faster than other specialized models like mTAND and Raindrop. The peak memory of the MissTSM+MAE (1.866 GB) is, however, slightly higher than the SAITS+MAE pipeline (0.74 GB), which can be further improved using more efficient code optimization in future works.
>
> > We have added this discussion on computational complexity to Appendix E.1 and E.2 in the revised manuscript.
>
> **Comment 3: Improve figure clarity: for bar charts, explicitly indicate whether lower or higher values are better, particularly for metrics other than F1 scores**
>
> > Thank you for this comment. We have updated the bar plot captions to explicitly indicate the metric information in our revised manuscript.

---

### Review · Reviewer_AS76 · 2025-11-05

**Summary Of Contributions:**

The paper proposes MissTSM, which is a model-agnostic and imputation-free layer for handling irregularly sampled multivariate time series. It combines Time-Feature Independent (TFI) embeddings, which treat each observed scalar as a token with 2D positional encoding, and a Missing Feature-Aware Attention (MFAA) mechanism that aggregates observed features at each time step via masked cross-attention. Integrated into transformer backbones, MissTSM achieves strong performance across forecasting and classification tasks, especially under high missingness (>70%).

Key strengths: elegant and practical design, broad empirical validation, strong robustness to missing data, model-agnostic applicability

Key weaknesses: potential information bottleneck from feature aggregation, limited scalability analysis, modest novelty of the attention mechanism

**Audience:**

Yes

**Audience Explanation:**

Researchers in time-series modeling, healthcare analytics, and machine learning for irregular data would find this work valuable, for a model-agnostic way to handle missing or irregularly sampled data.

**Broader Impact Concerns:**

No major ethical or societal concerns are apparent.

**Claims And Evidence:**

Yes

**Claims Explanation:**

The claims are well-supported through experiments across multiple datasets, masking schemes, and backbone models. They consistently demonstrate MissTSM’s robustness and competitiveness against both imputation-based and specialized IMTS methods. Results and ablations are clearly presented and reproducible.

**Requested Changes:**

1. Include an analysis of runtime and memory usage as $N$ increases, highlighting when the $T \times N$ tokenization becomes computationally limiting.

2. Expand the discussion on how compressing all variates into a single latent vector $L_t$ may constrain feature interaction modeling and suggest potential extensions.

3. It seems that the contribution lies in applying masked cross-attention to irregular time-series data rather than introducing a new attention mechanism. This should be emphasized in the paper.

---

> ### Author Response · Authors · 2025-11-13
> **Author Response to Reviewer AS76**
>
> **Comment 1: Discuss potential information bottleneck from feature aggregation:  Expand the discussion on how compressing all variates into a single latent vector $L_t$ may constrain feature interaction modeling and suggest potential extensions**
>
> > We agree with the reviewer that by applying the conventional framework of self-attention, we can capture cross-variate interactions that are currently missing in our framework (we learn attentions between variates and a learnable query vector at every time-step). We decided to learn a single query vector to reduce the computational cost of learning attention weights (we only need N x 1 attention weights instead of N x N, where N is the number of variates). While we empirically show that our simpler attention architecture still captures valuable information across variates, our work can be easily extended in future works to model all variate-pairs interactions, e.g., by introducing a hierarchical or grouped aggregation mechanism that first models variate dependencies for a single time-step and then performs a global pooling across time-steps.
>
> > We have added this discussion to Appendix D.4 in the revised manuscript.
>
> **Comment 2: Limited scalability analysis: Include an analysis of runtime and memory usage as N increases, highlighting when the T x N tokenization becomes computationally limiting**
>
> > To analyze the scalability of our $T \times N$ tokenization (where $T$ is number of time-steps, $N$ is number of variates), we conducted an empirical analysis of runtime and memory usage as $N$ increases. We generated multiple synthetic datasets using a combination of sinusoidal seasonality, linear trends, and random noise (around 56k observations) with a 70% MCAR missing fraction, by varying $N$ in the following range: $\{10, 20, 50, 100, 200, 500, 800, 1000\}$, while setting $T$ and embedding dimension $d$ to be 336 and 16, respectively. We implemented MissTSM with an iTransformer backbone and compare its performance with a baseline approach of using SAITS imputation with iTransformer in the Table below. We report the avg. time and peak GPU memory used per training epoch for both forward pass and forward + backward pass. We have also included a plot of these numbers in Appendix E.3 to make it easier to visualize trends in runtime and memory complexity as a function of $N$. Note that the reported baseline results reflect only the runtime and memory requirements of the iTransformer model itself, excluding the additional computational cost associated with the SAITS imputation process, for ease of analysis.
> | **N** | Forward Pass — Time (s) | Forward Pass — Time (s) | Forward Pass — Peak GPU (GB) | Forward Pass — Peak GPU (GB) | Forward+Backward — Time (s) | Forward+Backward — Time (s) | Forward+Backward — Peak GPU (GB) | Forward+Backward — Peak GPU (GB) |
> |:----:|:----------------------:|:----------------------:|:----------------------------:|:----------------------------:|:----------------------------:|:----------------------------:|:--------------------------------:|:--------------------------------:|
> |      | MissTSM | iTrans | MissTSM | iTrans | MissTSM+iTrans | iTrans | MissTSM+iTrans | iTrans |
> | 10   | 2.095   | 1.784  | 0.080   | 0.064  | 22.416  | 13.495 | 0.271  | 0.177 |
> | 20   | 2.954   | 1.911  | 0.149   | 0.114  | 27.312  | 16.275 | 0.410  | 0.236 |
> | 50   | 6.004   | 3.163  | 0.359   | 0.275  | 49.903  | 18.514 | 0.847  | 0.427 |
> | 100  | 10.227  | 12.487 | 0.711   | 0.576  | 96.618  | 48.387 | 2.764  | 1.925 |
> | 200  | 20.140  | 24.188 | 1.410   | 1.309  | 174.575 | 69.040 | 3.313  | 1.636 |
> | 500  | 52.917  | 62.014 | 3.515   | 4.536  | 456.296 | 185.875| 9.182  | 5.034 |
> | 800  | 83.468  | 113.758| 5.613   | 9.309  | 1204.278| 335.082| 17.028 | 10.403 |
>
> > We can see that both runtime and memory of MissTSM scales linearly during the forward pass, similar to the baseline approach. This trend is consistent with the theoretical analysis of MissTSM’s complexity that we have now added in Appendix E. However, the combined forward-backward runtime shows super-linear growth for MissTSM beyond $N$>100 compared to the baseline approach, which suggests that the training time is dominated by backpropagation step. On a single 24 GB NVIDIA RTX, training MissTSM becomes limiting at $N \approx 1000$. However, note that $N = 1000$ represents an extreme setup involving very large number of variates, while for common practical ranges of $N$ (e.g., $N < 50$), the computational overhead of backpropagation with MissTSM remains minimal. We have also included an empirical comparison of the computational performance of MissTSM with baseline IMTS models on the PhysioNet P19 dataset in Appendix E.2, where we show that MissTSM shows lowest training time, peak GPU memory, and highest inference throughput compared to baselines, while enjoying the lowest number of parameters and highest AUROC score.
>
> > We have added this discussion to Appendix E.3 in the revised manuscript.

---

> ### Author Response · Authors · 2025-11-13
> **Author Response to Reviewer AS76 (contd.)**
>
> **Comment 3: Modest novelty of the attention mechanism. It seems that the contribution lies in applying masked cross-attention to irregular time-series data rather than introducing a new attention mechanism**
>
> > We acknowledge that the main contribution of our work is not to introduce a new attention mechanism, rather, it lies in the novel adaptation of masked cross-attention for effectively modeling IMTS data, which to the best of our knowledge, has not been explored in any prior work.
>
> > We have added this clarification in Section 3.3 of the revised manuscript.

---

### Author Response · Authors · 2025-11-13
**Global Response to Review Comments**

We sincerely thank all reviewers for providing constructive feedback. We are encouraged that the reviewers found our work to present an elegant and practical model for modeling irregular multivariate time-series (IMTS) data, with broad applicability and strong empirical evaluation. In response to the reviewer comments, we have conducted several additional experiments and revised some of the discussions. Here is a summary of the changes:

1. We have included new theoretical and empirical analysis of MissTSM’s computational runtime and memory requirements in comparison with the baselines (*Reviewers bYD2, aMoN*). **Key insight:** *MissTSM+MAE framework trains 42x faster and achieves 44x higher inference throughput than the two-stage SAITS+MAE imputation pipeline* (*see Appendix E.1 and E.2 for more details*).

2. We have performed scalability analysis of MissTSM with respect to the number of variates (*Reviewer AS76*). **Key insight:** *MissTSM's computational cost scales linearly during the forward pass* (*see Appendix E.3 for more details*).

3. We have evaluated MissTSM using a non-transformer backbone, LSTM, and compiled comparisons of MissTSM with baseline methods on a common backbone. (*Reviewer bYD2*). **Key insight:** *MissTSM demonstrates competitive performance even with non-transformer backbones such as LSTM* (*see Appendix F.8.2 for more details*).

4. We have performed extensive hyper-parameter tuning of imputation methods and updated comparison of baselines with MissTSM (*Reviewer bYD2*). **Key insight:** *MissTSM maintains competitive performance despite improvements from tuning imputation methods* (*see Appendix F.6 for more details*).

5. We have added new discussions on MissTSM design choices (*Reviewer bYD2*) (*see Appendix D.4 for more details*).

6. We have added a Broader Impact Statement at the end of the Conclusion section (*Reviewer bYD2*).

All the corresponding results, discussions, and updates have been incorporated into the revised manuscript and are detailed below in response to each reviewer’s comment. The changes are colored red for better readability.

---

### Decision · Action_Editor_cFJi · 2025-12-10

**Recommendation:** Accept as is

**Audience:**

Yes

**Audience Explanation:**

Yes, the findings would be of significant interest to the TMLR audience. Irregularly-sampled multivariate time series is a pervasive challenge across many high-impact domains including healthcare, environmental monitoring, finance, and IoT applications. The paper addresses a practical pain point by offering a model-agnostic, imputation-free alternative that can be integrated as a plug-and-play layer with existing time series architectures, making it immediately useful to practitioners who want to leverage state-of-the-art backbones without committing to specialized IMTS architectures or complex two-stage imputation pipelines. The insights about when imputation-free approaches excel versus when simpler methods suffice provide actionable guidance for researchers and practitioners choosing among IMTS strategies. The demonstrated computational efficiency gains further enhance practical appeal, particularly for real-time or resource-constrained applications.

**Claims And Evidence:**

Yes

**Claims Explanation:**

Yes, the claims are well-supported by comprehensive empirical evidence. The authors validate MissTSM across multiple datasets, various missing value scenarios, different missingness fractions, and multiple backbone architectures. The experiments consistently demonstrate that MissTSM achieves competitive or best performance compared to both imputation-based pipelines and specialized IMTS methods. The revised manuscript strengthens the evidence further by adding computational analysis, scalability experiments with varying numbers of variates, hyperparameter tuning of baseline imputation methods for fairer comparison, and non-Transformer backbone experiments (LSTM) that validate the model-agnostic claims. Ablation studies confirm the value of individual components, and the results are reproducible through provided code and detailed implementation specifications in the appendix.